# DSCIM-Coastal v1.1: An Open-Source Modeling Platform for Global Impacts of Sea Level Rise

Nicholas Depsky[1,3,*], Ian Bolliger[2,3,4,*], Daniel Allen[3], Jun Ho Choi[7], Michael Delgado[2], Michael Greenstone[5,7], Ali Hamidi[2,4], Trevor Houser[2], Robert E. Kopp[6], and Solomon Hsiang[3,5]

[1]Energy & Resources Group, University of California, Berkeley, California, USA
[2]The Rhodium Group, Oakland, California, USA
[3]Global Policy Lab, Goldman School of Public Policy, University of California, Berkeley, California, USA
[4]BlackRock, San Francisco, California, USA
[5]National Bureau of Economic Research
[6]Department of Earth & Planetary Sciences and Rutgers Institute of Earth, Ocean and Atmospheric Sciences, Rutgers University, New Brunswick, New Jersey, USA
[7]Energy Policy Institute, University of Chicago, Illinois, USA
[*]These authors contributed equally to this work.

**Correspondence:** Nicholas Depsky (njdepsky@berkeley.edu), Ian Bolliger (ian.bolliger@blackrock.com)

**Abstract.**

Sea level rise (SLR) may impose substantial economic costs to coastal communities worldwide, but characterizing its global impact remains challenging because SLR costs depend heavily on natural characteristics and human investments at each location—including topography, the spatial distribution of assets, and local adaptation decisions. To date, several impact models have been developed to estimate the global costs of SLR. Yet, the limited availability of open-source and modular platforms that easily ingest up-to-date socioeconomic and physical data sources restricts the ability of existing systems to incorporate new insights transparently. In this paper, we present a modular, open-source platform designed to address this need, providing end-to-end transparency from global input data to a scalable least-cost optimization framework that estimates adaptation and net SLR costs for nearly 10,000 global coastline segments and administrative regions. Our approach accounts both for uncertainty in the magnitude of global mean sea level (GMSL) rise and spatial variability in local relative sea level rise. Using this platform, we evaluate costs across 230 possible socioeconomic and SLR trajectories in the 21st century. According to the latest Intergovernmental Panel on Climate Change Assessment Report (AR6), GMSL is likely to rise during the 21st century by 0.40-0.69 meters if late-century warming reaches $2°$ C and by 0.58-0.91 m with $4°$ C of warming (Fox-Kemper et al., 2021). With no forward-looking adaptation, we estimate that annual costs of sea level rise associated with a $2°$ scenario will likely fall between \$1.2 and \$4.0 trillion (0.1 and 1.2% of GDP, respectively) by 2100, depending on socioeconomic and sea level rise trajectories. Cost-effective, proactive adaptation would provide substantial benefits, lowering these values to between \$110 and \$530 billion (0.02 and 0.06%) under an optimal adaptation scenario. For the likely SLR trajectories associated with $4°$ C warming, these costs range from \$3.1 to \$6.9 trillion (0.3% and 2.0%) with no forward-looking adaptation and \$200 billion to \$750 billion (0.04 to 0.09%) under optimal adaptation. IPCC notes that deeply uncertain physical processes like marine ice cliff instability could drive substantially higher global sea level rise, potentially approaching 2.0 m by 2100 in very high emission

scenarios. Accordingly, we also model the impacts of 1.5 and 2.0 m GMSL rises by 2100; the associated annual cost estimates range from $11.2 to $30.6 trillion (1.2% and 7.6%) under no forward-looking adaptation and $420 billion to $1.5 trillion (0.08 to 0.20%) under optimal adaptation. Our modeling platform used to generate these estimates is publicly available in an effort to spur research collaboration and support decision-making, with segment-level physical and socioeconomic input characteristics

provided at https://doi.org/10.5281/zenodo.7693868 and model results at https://doi.org/10.5281/zenodo.7693869.

## 1   Introduction

Global mean sea level (GMSL) is projected to increase between 0.40-0.69 m for $2°$C of warming and 0.58-0.91 m for $4°$C of warming by 2100, though accelerated ice-sheet instability could result in substantially higher values (approaching 2 m) by end-of-century (Fox-Kemper et al., 2021). Coastal communities and ecosystems will experience a variety of impacts, including

more frequent tidal flooding, higher extreme sea levels (ESLs)[1], erosion, wetland degradation, salinization of soils and water reservoirs, and loss of land area to permanent inundation (Oppenheimer et al., 2019; Nicholls et al., 2006). The magnitude of relative sea level rise (RSLR) and associated impacts will vary by locality, depending upon global greenhouse gas (GHG) emissions (Fox-Kemper et al., 2021), ice sheet instabilities (DeConto et al., 2021; Bamber et al., 2019; Fox-Kemper et al., 2021), local atmosphere-ocean dynamics (Fox-Kemper et al., 2021), economic growth along coastlines (O'Neill et al., 2017;

Neumann et al., 2015; Armstrong et al., 2016), and adaptation actions (Hinkel et al., 2018; Diaz, 2016; Hinkel et al., 2014; Lincke and Hinkel, 2021).

    Despite advances in our understanding of GMSL, the global costs of these changes remain poorly constrained. A key obstacle to quantifying these global impacts is their strong dependence on the details of local conditions, such as topography, the spatial distribution of populations and assets, and local adaptation decisions. A challenge for modelers is the dual objectives of fully

accounting for these various factors at the local granularity necessary for accurate representation while also scaling these calculations globally. Improvements in computation and data availability now make achieving these two objectives feasible, but it has remained challenging for existing custom-built systems to be regularly updated to reflect new insights or improvements to global data sets describing local conditions.

    This paper presents what is to our knowledge the first fully open-source coastal modeling platform that (i) integrates up-to-

date local data on socioeconomic and physical conditions along coastlines globally, (ii) projects the physical, socioeconomic and ecological impacts of SLR along coastlines and (iii) directly models the costs and benefits of both retreat and protection as potential adaptation strategies. The platform is fully coded in the open-source computer language Python (v3.9) and integrates recently released, satellite-augmented global data layers describing coastal elevations, local sea levels, and the distribution of population and physical capital with widely used socioeconomic datasets. These data layers are projected onto 9,568 unique

coastal segments that span global coastlines. Each of these segments is then modeled as independently choosing across local, forward-looking adaptation strategies in an effort to minimize overall losses, following the framework developed in Diaz (2016). Using this platform, we evaluated net costs across 230 possible socioeconomic and SLR trajectories in the 21st century

---

[1]Terminology and acronyms for concepts related to sea level align with those recommended for contemporary use in Gregory et al. (2019).

to present here, though the tool is capable of accommodating tens to hundreds of thousands of future simulations in parallel if desired.

With no forward-looking adaptation, we estimate that annual global costs of sea level rise associated with $2°$ of warming (+0.40-0.69 m GMSL by 2100) will fall between \$1.2 and \$4.0 trillion (0.1% and 1.2% of GDP) by 2100, depending on socioeconomic and SLR trajectories. Locally cost-effective adaptation strategies could drastically lower these estimates to between \$110 and \$530 billion (0.02 and 0.06%). For the likely SLR trajectories associated with $4°$ C warming, these costs range from \$3.1 to \$6.9 trillion (0.3% and 2.0%) with no forward-looking adaptation and \$200 billion to \$750 billion (0.04

to 0.09%) under optimal adaptation. Under a very high emissions scenario with SLR projections that include the influence of deeply uncertain physical processes like marine ice cliff instability, end-of-century GMSL rise reaches +1.5-2.0 and the associated annual cost estimates range from \$11.2 to \$30.6 trillion (1.2% and 7.6%) with no forward-looking adaptation and \$420 billion to \$1.5 trillion (0.08 to 0.20%) under optimal adaptation.

All code used to aggregate and combine input data, as well as to estimate SLR impacts, is publicly available. This encourages

further development by the coastal impacts research community and modularizes the modeling process to facilitate seamless incorporation of future improvements to input datasets and additional model components.

## 1.1 The Basic Architecture of Global Coastal Impact Models

Global coastal models that estimate impacts of SLR and ESLs seek to quantify the exposure of some variable(s) of concern, such as human population, capital assets, and coastal ecosystems, to these physical hazards. They generally report the mag-

nitude of exposure to these hazards as their final output, and convert this exposure into some outcome of interest, such as economic losses (Hinkel et al., 2014; Diaz, 2016; Lincke and Hinkel, 2018). These models usually contain spatially explicit representations of physical coastline characteristics (e.g. coast lengths, elevation and land surface areas), exposure variables, and physical hazard variables.

To estimate future impacts, global coastal models must assume or model trajectories of pertinent physical and socioeconomic

values over time. Most climate change-oriented impacts models assess multiple trajectories of GMSL and many account for local RSLR and associated ESLs, which commonly correspond to different GHG emissions pathways (Hinkel et al., 2014; Diaz, 2016; Lincke and Hinkel, 2018, 2021). They may also contain different future trajectories of human population and capital asset growth, such as those represented in the Shared Socioeconomic Pathways (SSPs) database (Riahi et al., 2017; Hinkel et al., 2014; Lincke and Hinkel, 2018; Tiggeloven et al., 2020; Lincke and Hinkel, 2021).

The spatial and temporal resolution of model components can vary between studies and is sometimes limited by the resolution of available input datasets and/or by available computing resources. Additionally, many models also include some form of adaptive decision-making, such as allowing different coastal segments to construct protective coastal barriers (Hinkel et al., 2014; Diaz, 2016; Lincke and Hinkel, 2018; Tiggeloven et al., 2020; Lincke and Hinkel, 2021) or retreating inland (Diaz, 2016; Lincke and Hinkel, 2021), usually guided by some form of local cost-benefit analysis.

 **1.2 Closely Related Efforts and Platform Genealogy**

Several past studies employed global coastal impact models to estimate future damages from SLR and ESLs under various trajectories of global GHG emissions, socioeconomic scenarios, and adaptation pathways for thousands of sub-national coastline segments (Hinkel et al., 2014; Diaz, 2016; Lincke and Hinkel, 2018, 2021). Many of these studies used the Dynamic Interactive Vulnerability Assessment (DIVA) Coastal Database and modeling tool as their source of information for describing local coastlines. Originally developed by the Dynamic and Interactive Assessment of National, Regional and Global Vulnerability of Coastal Zones to Climate Change and Sea-Level Rise (DINAS-COAST) project (Vafeidis et al., 2008; Hinkel and Klein, 2009), the DIVA database partitions global coastlines into 12,148 segments and provides local physical attributes (e.g., inundation areas by elevation, extreme sea level heights, wetland areas, erosion characteristics) as well as socioeconomic characteristics (e.g. population densities, land use), allowing for spatially disaggregated coastal impact analyses (Vafeidis et al., 2008; Hinkel and Klein, 2009). At the time of its initial release in 2008, DIVA represented a substantial improvement over previous global, coastal databases and impact studies, which were most commonly performed using data at much coarser spatial resolutions (Hoozemans et al., 1993; Yohe and Tol, 2002; Nicholls, 2004, 2002; Dronkers et al., 1990; Pardaens et al., 2011; Hinkel et al., 2013). Presently, however, the DINAS-COAST program is no longer funded, and the accessibility of the DIVA database has fluctuated. Recently, a landing page has been created for the DIVA model at http://diva.globalclimateforum.org, though as of early 2023 the corresponding dataset is only available via direct correspondence with its authors. The underlying code and input data used to construct the DIVA database is not publicly available, making it difficult to replicate prior studies' results and diagnose issues that have appeared in previous versions of the dataset (Sect. 2.5.1). In this work, we address these issues of accessibility and transparency by generating a publically-available global dataset of coastal socioeconomic metrics, updating all core data layers used to generate DIVA and releasing the data assimilation model used to aggregate these into the final data product. The full set of data updates are described in Sect. 2 below.

In a key early analysis, Hinkel et al. (2014) employed the DIVA database to model the combination of coastal flood damages and adaptation (specifically, protective levee construction) under twelve scenarios of future RSLR and socioeconomic projections for sub-national coastal zones. Sea level rise scenarios in this study were constructed from estimates of global thermal expansion and regional ocean dynamic sea level data corresponding to low-, medium-, and high-emissions Coupled Model Intercomparison Project Phase 5 (CMIP5) experiments (Taylor et al., 2012) (Representative Concentration Pathways 2.6, 4.5, and 8.5) in four Earth System Models (ESMs), combined with low, medium, and high land-ice scenarios. The study also evaluated two different digital elevation models (DEMs) for estimating population exposure in coastal floodplains to SLR and ESLs, the GLOBE DEM (GLO, 1999), which was the original DEM used in DIVA (Hinkel and Klein, 2009), and the more recent Shuttle Radar Topography Mission (SRTM) DEM (Rodriguez et al., 2005). They found that their results were highly sensitive to the choice of DEM, which underscores the importance of updating global data layers used in coastal impact modeling as improved products are made available, which is one of the central aims of the work we present in this paper.

Expanding on the approach of Hinkel et al. (2014), Diaz (2016) developed the Coastal Impact and Adaptation Model (CIAM), a global modeling tool that estimated 21st century costs and adaptation strategies for each DIVA segment. One

core innovation presented in CIAM was that it allowed for each segment to choose between dike construction, as in Hinkel et al. (2014), and managed or reactive retreat. However, an obstacle to widespread usage of CIAM was its development in the commercial General Algebraic Modeling System (GAMS) closed-source platform. We build on the work by Diaz (2016), using the underlying decision-making framework of CIAM; however, we adapt, re-code, and optimize CIAM in the open-source Python computing language.

The architecture of CIAM was designed to capture key aspects of local adaptive decision-making that will likely be used by coastal communities worldwide. The objective of CIAM was to develop an optimization framework that could be applied locally, but generalized globally. To limit the computational challenge of solving stochastic dynamic programs for thousands of independent coastline segments, Diaz (2016) simplified the set of possible adaptation choices to a set of discrete decisions that are calibrated to local conditions. CIAM differentiated between six types of costs (i.e. "damages") due to RSLR and ESLs (Sect. 2.2): (a) the cost of permanent inundation of immobile capital or land, and ESL-related (b) damages to capital, (c) mortality, (d) expenditures on protection (i.e. infrastructure construction), (e) relocation costs, and (f) wetland loss. Possible protection actions include constructing levees at the 10, 100, 1000, and 10000-year ESL heights at each segment, and possible retreat actions include proactively vacating all land area under local mean sea level or within the 10, 100, 1000, or 10000-year ESL floodplain. Simulations in CIAM are implemented using discrete time-steps, termed "adaptation planning periods" (40-50 years), during which each segment updates their retreat or protection height based on the maximum RSLR projected to occur within the period. CIAM also allows for modelers to select a "no planned adaptation" option that constrains retreat to be reactive, rather than forward-looking, such that the population and capital assets only choose to relocate inland once they are permanently inundated by rising sea levels. Diaz (2016) considered a single socioeconomic growth trajectory based on the 2012 United Nations World Population Prospects (UN DESA, 2012), Penn World Table version 7.0 (Heston et al., 2011) and the 2011 IMF World Economic Outlook (IMF, 2011) projections and uses DIVA's older GLOBE DEM. The SLR trajectories used by Diaz (2016) were the 5th, 50th, and 95th percentiles of probabilistic RSLR projections from Kopp et al. (2014) for RCPs 2.6, 4.5, and 8.5, as well as a no-SLR baseline.

Here, we build on the approach of Diaz (2016), adapting and optimizing the decision-framework of CIAM to an entirely new set of global data inputs (i.e. replacing DIVA) and an open-source computer language. Given continued advancement in sea level rise modeling efforts and the improvement of global data inputs (e.g. coastal DEMs), it is essential that coastal impacts modeling platforms are able to integrate these updates. Additionally, we believe that these platforms should be developed in an open-source, transparent, and reproducible framework that will allow for increased collaboration and more rapid iteration amongst coastal impacts researchers, as has been done for modeling communities across numerous scientific disciplines (von Krogh and von Hippel, 2006). The platform we develop addresses these objectives by integrating the latest available physical, climate, and socioeconomic input data for an expanded suite of future SLR and economic growth trajectories in an updated and open-source version of the CIAM framework that, in addition to improved accessibility and transparency, results in greater resolution and substantially improved computational efficiency.

### 1.3 This Study: The Data-driven Spatial Climate Impact Model - Coastal Impacts

This modeling platform was developed as the sea level rise impacts module of the Data-driven Spatial Climate Impact Model (**DSCIM**) architecture (Rode et al., 2021; Carleton et al., 2022), and is thus named **DSCIM-Coastal**. It is partitioned into two distinct components (see Fig. 1), each made available as open-source products: (i) the collection, harmonization, and aggregation of updated physical and socioeconomic input datasets by coastal segment, which is named the Sea Level Impacts Input Dataset by Elevation, Region, and Scenario, or **SLIIDERS**, and (ii) the modeling platform itself, called **pyCIAM** (short for "Python-based CIAM"). Both components have been developed in accordance with FAIR Guiding Principles for scientific data management (Wilkinson et al., 2016) that are intended to improve the Findability, Accessibility, Interoperability, and Reuse of scientific data.[2]

The SLIIDERS data set is conceptually similar to DIVA in that it contains a suite of variables defined across a collection of coastal segments designed for coastal impact modeling efforts. However, while DIVA is not publicly accessible, SLIIDERS and all of its components are available with open access licenses, thereby supporting transparency and replicability of coastal damage analyses for research communities around the globe.[3] In addition, the partition of global coastlines that defines separate coastal segments as units of analysis has been revamped in order to achieve greater balance in geographic coverage and reduce redundant computations. SLIIDERS also contains updated topographic, geographic, and socioeconomic input datasets, including refined coastal DEMs and socioeconomic growth trajectories.

pyCIAM is an open-source, computationally efficient and functional modeling platform for segment-level adaptation decision making that incorporates the following improvements to the original implementation of CIAM (Diaz, 2016): (i) updates to (and expansion of) all topographic, geographic and socioeconomic input data using SLIIDERS, and updated oceanographic inputs using a large suite of 23 SLR projections, (ii) improvements to model representation of different variables, such as population and capital asset distribution and storm damage calculations, (iii) availability as an open-source, self-contained Python package and input database, making the workflow easily accessible and modifiable for other researchers, and (iv) improved computational efficiency and scalability, enabling the application of CIAM to large, probabilistic ensembles of sea level change.

The pyCIAM model is configured to utilize the SLIIDERS inputs and SLR projections presented here, but can easily be run using a modified set of inputs or SLR pathways, provided the data structure is consistent with this configuration. Similarly, the SLIIDERS product can be used independently from pyCIAM as inputs for other coastal analysis or as contextual information

---

[2]These data and modeling components abide by the FAIR criterion as specified by The Future of Research Communications and e-Scholarship (FORCE11). Specifically, they are i) Findable via unique and persistent identifiers, with these identifiers specified in component metadata and indexed in a searchable resource (Zenodo, Github); ii) Accessible in that they are retrievable via these identifiers and are open, free and universally implementable; iii) Interpretable through the use of a formal, accessible, shared and broadly applicable language/vocabulary (manuscript and metadata in standard English and code in Python) and the inclusion of appropriate references to other data where necessary (e.g. input data sources); and iv) Reusable by specifying accurate and relevant attributes, applying an accessible data usage license and complying with coastal modeling community standards of language and data/code provision (Force11.org).

[3]One of the input data sources used in generating SLIIDERS, CoastalDEM (Kulp and Strauss, 2019), is not freely available at the resolution employed in this study but is available for research use at a lower resolution.

on coastal zones. It can also be recreated using alternate input sources as desired, as the scripts to generate the product are
provided with it.

The following sections describe how SLIIDERS and pyCIAM are constructed, show example results of model outputs and
diagnostics from 2005-2100 and compare to the results of Diaz (2016), and discuss current limitations to the model and input
datasets, outlining planned improvements and future research priorities.

## DSCIM-Coastal
Open-source platform for computing global coastal impacts as part of the Climate Impact Lab's multi-sectoral Data-driven Spatial Climate
Impact Model (DSCIM)

### Coastal Characteristics Dataset
Sea Level Impacts Input Dataset by Elevation, Region,
and Scenario for each coastal segment (**SLIIDERS**)

**Model Inputs**

### SLR Impact Modeling Platform

#### SLIIDERS

**Physical Variables**
- Segment location and coastline length
- Land area by elevation (0.1m elevation bins)
- Extreme sea levels (10, 100, 1000, 10000-year)
- Wetland and mangrove area (0.1m elevation bins)

**Socioeconomic Variables**
Present (2019) and projected values by SSP-IAM:
- Population
- GDP (annual per capita income)
- Physical capital
- Construction costs

#### pyCIAM
Python-based Coastal Impacts and Adaptation Model

**Least-cost adaptation option for each segment**
One of the following:
- protect to a given extreme sea level (ESL) height
- retreat proactively to a given ESL height
- retreat reactively to local relative sea level rise (RSLR) alone

**Segment-wise costs (i.e. damages) outputs**
Under least-cost option and for reactive retreat only:
- Permanent inundation of land due to LSLR
- Wetland/mangrove loss due to LSLR
- Capital stock damage due to ESLs
- Population mortality due to ESLs
- Relocation (reactive and proactive retreat)
- Protective barrier construction

#### Sea Level Rise Projections
Local relative SLR projections from 23 distinct
scenarios for different emissions pathways:
- IPCC Sixth Assessment Report ($n=8$)
  - 5 medium-confidence, 3 low-confidence
- U.S. Inter-Agency SLR Technical Report ($n=5$)
- IPCC Fifth Assessment Report ($n=3$)
- IPCC SROCC ($n=3$)
- Increased ice sheet instability scenarios ($n=5$)

**Figure 1.** Components of the Coastal portion of the Data-driven Spatial Climate Impact Model (DSCIM-Coastal)

## 2  Methods and Data

We constructed the Python Coastal Impacts and Adaptation Model (pyCIAM) by adapting the original code and structure of the
Coastal Impacts and Adaptation Model (CIAM) (Diaz, 2016), obtained from http://github.com/delavane/CIAM in June 2020,
with changes subsequently made in three phases:

1. Porting the model from GAMS to a standalone Python module (creating *pyCIAM*)

2. Updating all model inputs with the *SLIIDERS* data and SLR projections, constituting newer, improved physical and socioeconomic datasets

3. Implementing changes to the model functionality itself for the purposes of:
   – Computational efficiency
   – Updating assumptions where new data provided previously unavailable insights
   – Aligning model implementation with the model description in Diaz (2016)
   – Reducing noise in numerical approximation algorithms

## 2.1 Model Structure

The aspects of CIAM as presented in (Diaz, 2016) that are maintained in pyCIAM include the segment-based structure of the model and the adaptation actions that each segment is permitted to take throughout the modeling period, comprised of the following options:

– *Reactive Retreat*: When a portion of land falls below MSL, all people and mobile capital are relocated to an unaffected, inland region away from the coast that is not in danger of future impacts from SLR or ESLs, and immobile capital is abandoned.

– *Protection*: Construction of a generic levee to protect the entire coastline segment. Available choices for protection height include the 10, 100, 1000, and 10,000-year return values of ESL. This height changes linearly with RSLR.

– *Proactive Retreat*: All people and mobile capital below a certain retreat height are assumed to be relocated to a safe, inland region, and immobile capital below that height is abandoned. The options for that retreat height level are discretized to the same values available for protection, with the addition of a "low retreat" option representing the maximum MSL projected during a "planning period" (10 years).

Note that, as described in Diaz (2016), each coastal segment may only choose one adaptation option, e.g. *retreat-1000*, for the entire model duration. While the height of the retreat level changes over time as the 1000-year ESL return value changes due to RSLR, the segment cannot, for example, choose retreat-100 for the first 40 years and then protect-10000.

The model is discretized into time steps (10 years in the original CIAM, annual in pyCIAM), during which all time evolving parameters are held constant. In addition, the segments use a configurable set of "planning periods" (40-50 years each period in CIAM, 10 years each in pyCIAM), which each correspond to a set of one or more timesteps. For each planning period, a single height is chosen for retreat or protection (assuming the segment does not select "reactive retreat") that represents the maximum height projected for the chosen ESL return value during the planning period.

## 2.2 Cost Calculation

Following Diaz (2016), pyCIAM separately tracks inundation costs, retreat costs, protection costs, cost of wetlands loss, and extreme sea level damage and mortality. These categories of costs are all used in cost minimization, and each is detailed below.

### 2.2.1 Inundation Costs

This category reflects the value of land and immobile capital lost to inundation. In Diaz (2016), immobile capital was allowed to fully depreciate if the strategy chosen is proactive retreat, such that capital-related losses due to inundation are always 0. This was based on a theoretical argument that for a planned retreat, a rational social planner would cease the creation of new physical capital far enough in advance that all remaining capital would have fully depreciated by the time the retreat occurs (Yohe et al., 1995). However, this assumption has been critiqued in subsequent work (Lincke and Hinkel, 2021) due to its lack of empirical grounding. Furthermore, it ignores the welfare loss associated with not replacing depreciating assets in the years leading up to retreat. These new capital investments would have been made in the absence of SLR, and thus the lack of investment should be counted when assessing total SLR impacts. Therefore, pyCIAM alters CIAM's assumption of full depreciation, instead modeling immobile capital to experience no excess depreciation beyond the background rate implicitly included in the capital growth model used to generate SSP-aligned capital projections. This results in the full estimated value of capital being lost when abandoned or inundated, in line with the assumptions of Lincke and Hinkel (2021).

Value of land lost permanently to inundation is estimated in accordance with the Diaz (2016) methodology, which approximates land values based on country-level assumptions of non-coastal land value from the integrated assessment model FUND (Tol, 1996). We assume that these national land values appreciate over time as a function of projected per capita income and population density growth in future years for each country. Equation 7 of the Diaz (2016) Supplemental Information details the total cost of inundation as a function of land values and immobile capital loss. Because pyCIAM, like CIAM, estimates land to have value even in unpopulated regions, non-zero inundation costs are still incurred in unpopulated segments due to lost land (or lost wetland area, see Sec. 2.2.4), despite the absence of any immobile capital losses. As expected, the magnitudes of these inundation losses tend to be much lower than in highly populated segments exposed to SLR.

### 2.2.2 Retreat Costs

This category reflects the costs of relocating population and mobile capital and of demolishing immobile capital. Following Diaz (2016), capital relocation costs are valued at 10% of total value, and immobile capital demolition costs are valued at 5%. In Diaz (2016), the intangible relocation cost is valued at one year of per capita income, which varies by country and over time. As described in the Diaz (2016) Supplemental Information, this was an arbitrary value chosen because it lay between the value used in FUND (3) and a number derived from personal communication with Robert Mendelsohn (0.5). We update this value to eight times local income, based on analysis described below in Sect. 2.3.

### 2.2.3 Protection Costs

This category reflects the construction and maintenance costs of building a protective levee, along with the value of lost land. As in Diaz (2016), maintenance costs are assumed to be 2% of the initial construction cost, and the value of lost land is calculated as the local land value (which varies over countries and years) times the length and width of the barrier, assuming a $60°$ slope.

### 2.2.4  Wetlands Loss

This category reflects the value of wetlands lost to either SLR or protection. As in Diaz (2016), wetlands are assumed to be able to partially absorb SLR up to 1 cm year$^{-1}$, with the degree of loss increasing quadratically with the rate of SLR. Above the critical threshold of 1 cm year$^{-1}$, all inundated wetlands are lost. In addition, all wetland area below a protective barrier is also assumed to be lost. More details on the calculation of wetland loss can be found in Equation 8 of the Diaz (2016) supplemental information.

### 2.2.5  Extreme Sea Level Capital Damage

This category reflects the value of capital loss occurring due to ESL events, using a depth-damage relationship that takes the shape $\frac{d}{1+d}$. The probability density function of ESL values at each segment location is represented as a Gumbel distribution, derived from Muis et al. (2016) in Diaz (2016) and from Muis et al. (2020) in pyCIAM. The product of this PDF and the estimated capital loss conditional on each ESL height in the distribution is integrated to obtain the annual expectation of ESL-driven capital loss per elevation slice, and these costs are summed over elevation to obtain the annual damages per segment (see Diaz (2016), Supplementary Material Section 2.1, Eqs 9-12). For computational efficiency, this set of discrete products, integrations, and sums is performed on a variety of example inputs prior to executing the actual CIAM model. In Diaz (2016), functions are fit to these outputs to relate ESL height to loss for different adaptation options, with unique coefficients for each segment.

$$D_{r,s,t} = (1 - \rho_{s,t})C_{s,t}\left(\frac{\sigma_{0,r,s}}{1 + \sigma_{A,r,s}exp\left(\sigma_{B,r,s}H_{r,s,t}\right)}\right) \tag{1}$$

$$D_{p,s,t} = (1 - \rho_{s,t})C_{s,t}\left(\frac{\sigma_{0,p,s} + \sigma_{1,p,s}S_{s,t}}{1 + \sigma_{A,p,s}exp\left(\sigma_{B,p,s}H_{p,s,t}\right)}\right), \tag{2}$$

where

$D_{r/p,s,t}$  is the ESL-driven expected capital loss conditional on retreat or protection to height $r$ or $p$, respectively, for segment $s$ in time step $t$,

$\rho_{s,t}$  is a country-level resilience factor (defined in Diaz, 2016),

$C_{s,t}$  is the capital density (in \$ per km$^2$),

$H_{r/p,s,t}$  is the difference between the retreat or protection height and local mean sea level,

$S_{s,t}$  is the local mean sea level, and

$\sigma$  are the fitted coefficients.

However, this has two notable issues. First, this fixed functional form may not fully represent heterogeneous relationships between adaptation height, MSL, and damage across segments, due to differing elevational distributions of capital at each segment. Second, in Diaz (2016) the damages conditional on a given retreat standard (e.g. 1-in-10 year ESL height) are a function only of the difference between MSL and the retreat standard, not of the absolute MSL height. This approximation

would be accurate if the same amount of capital exists at all elevations, independent of the area of land available at those elevations; however, elsewhere in the original CIAM model it is assumed that the elevation distribution of capital follows that of land area.

In pyCIAM, we address these issues by employing a multi-dimensional lookup table instead of these two functions. For each segment, we find the lowest and highest values of MSL ($S$) and of the difference between retreat/protection height and MSL ($H$) across all SLR scenarios we wish to simulate, all adaptation choices, and all timesteps. We then choose 100 equally spaced values between these bounds for each of the two variables. For both of the adaptation categories (retreat and protection), we now have 10,000 scenarios reflecting different combinations of $H$ and $S$. We normalize capital stock so that it sums to one, yielding fractional capital stock in each elevation slice. The current implementation assumes that these ratios remain fixed over time. However, should one wish to model within-country migration due to considerations such as SSP-consistent coastal urbanization and migration flows (e.g. Jones and O'Neill (2016); Merkens et al. (2016)), such changes can be accommodated by updating the appropriate variables in the SLIIDERS input dataset. For each of the 20,000 scenarios, we calculate damages using a discrete double integral over ESL height and elevation slice. In the pyCIAM model, the equations for damage are thus:

$$D_{r/p,s,t} = (1 - \rho_{s,t})K_{s,t}\gamma(H_{r/p,s,t}, S_{s,t}) \tag{3}$$

where

$D_{r/p,s,t}$     is the ESL-driven expected capital loss conditional on retreat or protection to height $r$ or $p$, respectively, for segment $s$ in time step $t$,

$\rho_{s,t}$     is a country-level resilience factor (defined in Diaz, 2016),

$K_{s,t}$     is the total value of capital stock in segment $s$ at time $t$,

$H_{r/p,s,t}$     is the difference between the retreat or protection height and local mean sea level,

$S_{s,t}$     is the local mean sea level, and

$\gamma$     is the bilinear interpolation function across $H$ and $S$, using the previously defined lookup table.

### 2.2.6 Extreme Sea Level Mortality

This category reflects the expectation of annual costs of mortality occurring due to ESL events, where death equivalents are valued using a Value of a Statistical Life (VSL) framework, as employed in Diaz (2016), which assumes 1% mortality for all populations exposed to a given ESL, based on Jonkman and Vrijling (2008). This is modeled similarly to the ESL-driven capital loss, except that the 1% mortality assumption is used in place of the depth-damage function. In the implementation of Diaz (2016), both the mortality assumption and the depth-damage function appear to have been used in conjunction, although the text of the Diaz (2016) paper states that the depth-damage function should only be used in the estimation of capital stock damage, not mortality. We therefore corrected this discrepancy in our implementation of ESL-driven mortality estimates in pyCIAM.

### 2.2.7 Least Cost Optimization

For each planning period, every segment considers each of the possible adaptation options and assesses costs at each annual time step within the period. Following Diaz (2016), we maintain the assumption that these decision-making agents have perfect foresight of projected RSLR over this planning period; however, we reduce these periods from 40-50 years to 10 years (Sect. 2.7.2). The maximum heights of projected RSLR at each segment during a given planning period in turn influence the heights at which protect or retreat adaptation options are employed. For segments that adapt via reactive retreat, the height of retreat exactly matches this projected RSLR, while segments employing 10, 100, 1000, 10000-year retreat or protect actions consider the heights of these ESLs atop this projected RSLR baseline for that planning period. Once adaptation costs are calculated for all adaptation periods, we follow (Diaz, 2016) and calculate the NPV across the entire model duration for each adaptation option, and each segment chooses the least cost option.[4]

### 2.3 Estimating Non-market Costs of Relocation

In pyCIAM we introduce a calibration of non-market retreat costs based on observed patterns of settlement. Non-market retreat costs are those costs that are not directly visible to the market, but which nonetheless are incurred by individuals if they chose to relocate. For example, the non-pecuniary emotional cost associated with moving or the loss of social networks due to moving would both be non-market retreat costs. Accounting for these impacts would indicate that the total welfare impact of forced relocation is greater than simply the market costs associated with abandoning immobile capital. The existence of non-market relocation costs are thought to explain the observation that some patterns of coastal adaptation currently would not appear to be economically rational based on market costs alone (McNamara et al., 2015; Armstrong et al., 2016; Haer et al., 2017; Bakkensen et al., 2018; Hinkel et al., 2018; Suckall et al., 2018). Using only market costs, least-cost optimization would indicate that many real-world populations should relocate or protect themselves, thus there must exist unobserved non-market costs that keep those populations in their current locations. We leverage this observation to estimate the approximate magnitude of non-market relocation costs that would be necessary to explain current global settlement patterns.

Though CIAM does include some non-market costs associated with moving, equivalent to one year of GDP, the model does not re-create observed patterns of settlement when it is initialized and run under an optimal adaptation scenario. Instead, it results in an excess of instantaneous relocation in the first period of the model run. This indicates that the non-market costs specified are likely too small, because they are insufficient to hold populations to their observed present locations before any SLR occurs in the model. Specifically, when the optimal adaptation scenario is run under the baseline parameterization in Diaz (2016) and with the assumption of no climate-driven sea level rise, we observe that $1.26T of capital and 33M people instantly relocate. Adjusting for population and capital growth over the century, this instant relocation represents 41% and 44% of the *cumulative* relocation realized by the end of the century under the median SLR scenario for RCP 4.5. This instantaneous relocation conflicts with the distribution of people and capital observed in the world today and suggests that there are larger costs of relocation than are accounted for in the original parameterization of CIAM used in Diaz (2016).

---

[4]In contrast to Diaz (2016), we include initial adaptation costs from the first planning period in this NPV calculation (Sect. 2.7.3).

The original parameterization of CIAM in Diaz (2016) assumed that non-market costs are equal in value to consumption of one year of local GDP per capita, based on this value falling between two alternative estimates: 0.5 years (obtained from the author's personal communication with Robert Mendelsohn) and 3.0 years, the value assumed in the FUND Integrated Assessment Model (Tol, 1996). Notably, the more recent evolution of FUND – the GIVE model (Rennert et al., 2022) – relies directly on CIAM for estimating costs of SLR and thus now assumes costs equivalent to one year of local GDP per capita. In a similar modeling framework to CIAM, Lincke and Hinkel (2021) used the FUND value directly and further provided a literature review that finds empirical and theoretical estimates of total relocation costs varying between 2.3 and 9.5 years of average local income per capita. These findings suggest that the factor of one used in Diaz (2016) may underestimate relocation costs.

To address this, we adopt an approach to calibrate these unobserved non-market costs of relocation against real world behavior. Our calibration approximates a "revealed preference" approach, in which the behavior of agents is thought to reveal information about their preferences and values that is not otherwise visible (other elements of DSCIM adopt related methods to estimate the undocumented costs of adaptation decisions in other sectors, e.g. see Carleton et al. (2022)). Intuitively, this strategy relies on the insight that if individuals found the benefits of moving to be larger than the combined market and non-market costs, they would relocate. We cannot observe the non-market costs, but we can estimate the benefits and the market costs. If we observe that individuals have not relocated but CIAM computes that the benefits outweigh the market costs even before considering SLR, then we can estimate a lower bound on the implied non-market costs (equal to the benefits minus the market costs) that must be present in order to prevent them from relocating and rationalize their observed behavior.

Our ability to recover non-market costs using a revealed preference approach is constrained by our ability to accurately model benefits and market costs of relocation. There are inherent limitations in a global model (e.g. input data inaccuracies, preference heterogeneity) such that, at a segment-level, there will likely be some segments where benefits and/or market relocation costs are not measured exactly. Thus, we choose a relocation cost parameter by taking the exposure-weighted median value of segment-specific estimates of non-market costs.

To do this, we identify the total population and physical capital that would instantaneously relocate when the model is initialized in the absence of non-market relocation costs, assuming median estimates of RSLR in a no-climate change scenario (i.e., no change in GMSL, and RSLR associated only with land subsidence). For this simulation, we choose middle-of-the-road socioeconomic projections characterized by SSP2 and the International Institute for Applied Systems Analysis (IIASA) GDP growth model (Crespo Cuaresma, 2017). We then steadily increase the relocation cost parameter until 50% of this population and capital no longer instantaneously relocates under the optimal adaptation scenario. This median approach balances the desire to capture the non-market costs causing observed non-relocation with the recognition that data and parameter limitations associated with a global model will inevitably cause some discrepancy between modeled and observed behavior. Because this median occurs at different values for population and physical capital, we average the two values (6.7 and 10.9 years of local income, respectively) to obtain the 8.0 factor used in pyCIAM. Fig. 2 illustrates this calculation.

We note that this approach is facilitated by the resolution of the input data represented in SLIIDERS. The DIVA inputs used in Diaz (2016) assume that population and capital density are homogeneously distributed throughout each segment, and are non-

varying by elevation. This both distorts the elevation distribution of the observed present-day state of these two variables and

370    prohibits the analysis described above. By leveraging global gridded datasets of population, capital, and elevation, SLIIDERS and pyCIAM capture heterogeneous density and better represent the true present-day elevation distribution of population and capital within each segment (Sects. 2.6.1, 2.6.3, and 2.5.3).

After updating the non-market relocation cost parameter, we additionally follow the approach of Lincke and Hinkel (2021) and do not distinguish between the non-market costs of reactive and proactive retreat. Diaz (2016) assigns five times higher

375    costs to reactive retreat, though there is no empirical basis reported for this additional cost. Thus, we assume that both proactive and reactive retreat in pyCIAM incur losses equivalent to 8.0 years of income, rather than one and five years, respectively, in Diaz (2016).

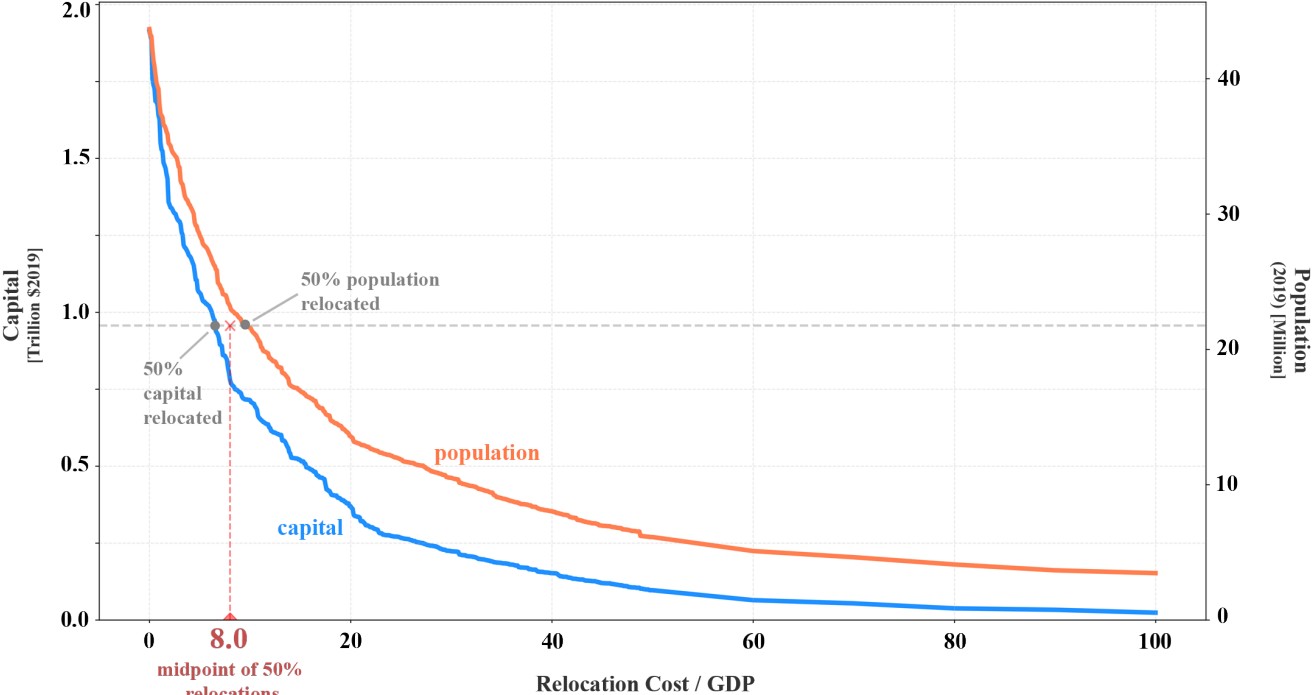

**Figure 2.** Calibrating the non-market relocation cost parameter based on the revealed preference of current populations. Curves show the magnitude of the population (orange) and physical capital (blue) that is instantaneously relocated in the optimal adaptation scenario of pyCIAM, assuming SSP2-IIASA socioeconomic projections and median no-climate change RSLR, as a function of this parameter. The parameter is normalized by local GDP per capita. We identify the parameter values for which 50% of the population and capital instantaneously relocated under an assumption of zero non-market costs are no longer relocated, and average these two values to estimate the relocation parameter used in pyCIAM.

## 2.4 Porting CIAM from GAMS to Python

CIAM was constructed in the closed-source General Algebraic Modeling System (GAMS) language. However, the model does not require the dynamic programming capabilities offered by GAMS. Therefore, porting the model to Python, a commonly used, open-source programming language, offers greater flexibility, access, and efficiency without loss of functionality. Before adding additional resolution to the model, pyCIAM computed a global run of a single SLR trajectory in 15-20 seconds, compared to 6-8 hours for CIAM. To ensure that this first stage of changes did not introduce changes to model functionality, we ensured that this version of pyCIAM replicated the results from the CIAM (in GAMS) model obtained from its source repository before updating model inputs. This replication was largely confirmed, with only very minor deviations between the computed results and those reported in (Diaz, 2016). The observed deviations were also reflected in the outputs of the unaltered CIAM model we obtained, suggesting that the configuration of the publicly available CIAM model was likely slightly altered from that used in Diaz (2016) (Table 1).

|  | Diaz 2016 (GAMS, reported in original paper) | CIAM (GAMS, computed in this study) | pyCIAM (Python, computed in this study) |
|---|---|---|---|
| Global NPV (2010-2100) | 1700 | 1692.2 | 1692.2 |
| U.S. NPV (2010-2100) | 419 | 419.7 | 419.7 |
| Australia NPV (2010-2100) | 208 | 208.6 | 208.6 |
| Brazil NPV (2010-2100) | 98 | 97.5 | 97.5 |
| China NPV (2010-2100) | 87 | 87.0 | 87.0 |
| Wetland Loss in 2100 | 80 | 79.3 | 79.3 |
| Global Costs in 2100 (optimal adaptation) | 270 | 282.1 | 282.1 |
| Global Costs in 2100 (no adaptation) | 2200 | 2251.5 | 2251.5 |
| Calculation runtime | - | 6-8 hours | 15-20 seconds |

**Table 1.** Comparison of select model estimated costs (in $2010B USD) as reported in Diaz (2016) with those calculated from the original CIAM code in GAMS obtained from its online source repository and those calculated by pyCIAM after porting CIAM to Python and before any additional changes. Values reflect median relative sea level rise projections from Kopp et al. (2014) under a high emissions scenario (RCP 8.5). Estimates also reflect total coastal costs. In other words, costs from a baseline "no climate change" scenario, including only background local relative sea level changes unrelated to changing global sea level, have not been subtracted. Model runs were conducted on an Apple MacBook Pro laptop with a 2.8 GHz Quad-Core Intel Core i7 processor and 16GB of RAM.

## 2.5 Physical Model Inputs in SLIIDERS

### 2.5.1 Coastal Segments

To improve the traceability of data inputs and the efficiency of model optimization, we replaced the irregular DIVA coastal segments with segments based on the points at which ESLs are estimated in the Coastal Dataset for the Evaluation of Climate Impacts (CoDEC). This represents a roughly uniform, 50-km spacing of global coastline points (Muis et al., 2020). We made a number of slight alterations to the original CoDEC point set and used these points as midpoints of 50-km coastline segments (Sect. A). The alterations ensured that (a) the coastline segments were nested by country boundaries, as the DIVA segments are, and (b) any extra points corresponding to offshore buoy gauges (used for validation in CoDEC) were removed. We also thinned European CoDEC points, originally provided at an extra fine 10km spacing, to 50km in order to have globally uniform spacing. In addition, we manually added 19 segments for small island states or small slivers of national coastlines not represented in the original CoDEC point set (e.g. Anguilla, Tokelau, Jordan's small coastline, etc.). The final subset of CoDEC coastal points utilized in pyCIAM totaled 9,568. Natural Earth coastlines were used to make the point-to-segment conversion (1:10m resolution[5]). The coastline lengths of each segment, used to calculate the potential costs of building protective barriers, were derived from this final set of segments (Sect. A1).

The decision to replace the coastal segments was motivated by several reasons. First, in the version of DIVA (v1.5.5) used in Diaz (2016), we found that many of the coastal segment lengths in high latitude regions were substantially overestimated, likely due to a geographic projection error. This error appeared to be corrected in versions of DIVA used in subsequent studies; however, we nevertheless wished to avoid dependence of pyCIAM on DIVA, with uncertainty surrounding its ongoing development support and dataset availability. Second, we found that DIVA contains a substantial over-representation of small, mostly unpopulated land masses in island regions within its set of 12,148 segments. For example, DIVA contains 1,316 individual segments for French Polynesia, constituting 10.8% of all global segments but representing less than 0.004% of global population. This created substantial computational inefficiencies, as all segments require roughly equivalent computation.

---

[5]The '1:10m' label indicates the scale of the physical vector layers, which can also be thought of as the maximum length of coastline across which simplification of complex coastlines into straight line segments can occur. 1:10m coastlines are the most granular product provided by Natural Earth.

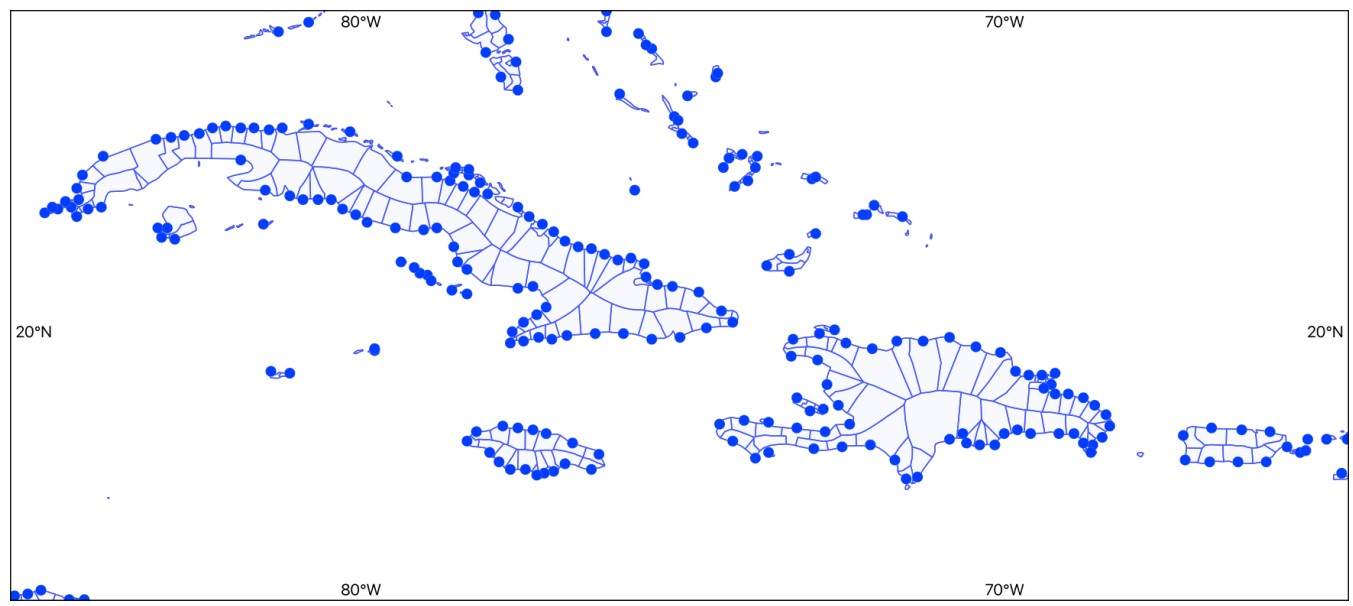

**Figure 3.** Example pyCIAM coastal segments (areas) and their centroids (points) for a region within the Caribbean.

### 2.5.2 Extreme Sea Levels

We obtained ESL distributions from CoDEC v1 (https://doi.org/10.5281/zenodo.3660927), which uses the third generation Global Tide and Surge Model (GTSM) combined with the ERA5 reanalysis to create a reanalysis product of historical sea levels (Muis et al., 2020). The CoDEC data provide the location and scale parameters of a Gumbel extreme value distribution fit to modeled ESLs at each coastline point, which we used to obtain the return periods required by CIAM (1, 10, 100, 1000, 10000-year). In validation analysis that compares CoDEC to observed tide gauge values, CoDEC values slightly underestimate annual ESL maxima by an average of 0.04m across all observed tide gauge stations, with 1-in-10 year mean ESL heights underestimated by 0.10m. Certain areas exhibit greater model bias, with 25% of tide gauge stations included in the validation showing absolute biases greater than 0.2m and 0.3m for annual and decadal maxima, respectively. In regions with a large tidal range and/or frequent tropical cyclones, biases are generally larger. See Muis et al. (2020) for a full discussion of CoDEC model validity.

### 2.5.3 Elevation

The use of accurate elevation data is crucial to appropriately representing sea level rise impacts (Kulp and Strauss, 2019). We have implemented an updated elevation model used to define the population and physical capital exposed to SLR in pyCIAM in the following manner.

1. We utilize the CoastalDEM v2.1 dataset (Kulp and Strauss, 2021) to define elevations at 1 arc-second resolution (roughly 30m). The v2.1 release of CoastalDEM represents further improvements to the initially-released product (v1.1) (Kulp

and Strauss, 2018), though both datasets represent substantial accuracy improvements to prior DEMs, such as the widely used SRTM DEM. In addition to higher resolution elevation estimates compared to the 30-arc-second GLOBE DEM used in Diaz (2016), CoastalDEM significantly reduces bias found in SRTM, as presented in a comparative analysis based on CoastalDEM's initial release (v1.1) (Kulp and Strauss, 2019). Compared to SRTM, CoastalDEM v1.1 suggests that roughly three times the amount of present day population resides below projected high tide levels under low emissions sea level rise scenarios by 2100 globally (Kulp and Strauss, 2019). It should be noted that the high-resolution version of CoastalDEM v2.1 is the only input used in this study that is not publicly available. It is obtained via license with Climate Central, the developers of the DEM, though lower-resolution versions of the dataset are freely available for academic use. For the small number of regions that we model where CoastalDEM does not exist (e.g. above and below 60N and 60S, respectively), we derive elevations from the SRTM15+ v2.5 dataset (Tozer et al., 2019).

2. We pair this DEM with 30 arc-second population estimates (LandScan 2021, Sims et al. (2022)) and capital stock (LitPop (Eberenz et al., 2020)) rasters, which allows for independent calculations of the distribution of land area, capital, and population with respect to elevation. We also rescale LitPop at the country-level to match more recently available data from Penn World Table 10.0 (Feenstra et al., 2015) and other sources (see Section 2.6.3). This approach differs from that of Diaz (2016), where population and capital stock densities were defined at the segment level and assumed to be homogeneously distributed within a segment.

3. We discretize the distributions of population and capital to 0.1m elevation slices, rather than 1.0m.

4. We mask all pixels that are not hydraulically connected to the ocean at 20 meters of SLR from analysis. This screens out most inland low-elevation areas not exposed to SLR. 20 meters is the highest elevation bin that we consider, reflecting the upper end of the ESLs that we consider combined with the upper end of local RSLR.

### 2.5.4 Wetlands and Mangroves

For wetland areas, pyCIAM utilizes the European Space Agency's GLOBCOVER v2.3 global land cover dataset from 2009, offered at a 300m resolution (obtained in May 2021 from http://due.esrin.esa.int/page_globcover.php) (European Space Agency and UCLouvain, 2010). Three different land cover classifications from this layer, as defined in (Hu et al., 2017), were coded as "wetlands":

1. Closed to open ($> 15\%$) broadleaved forest regularly flooded (semi-permanently or temporarily) - Fresh or brackish water

2. Closed ($> 40\%$) broadleaved forest or shrubland permanently flooded - Saline or brackish water

3. Closed to open ($>15\%$) grassland or woody vegetation on regularly flooded or waterlogged soil - Fresh, brackish or saline water

Mangrove extents were updated using values from UNEP's Global Mangrove Watch 2016 dataset (Bunting et al., 2018) (obtained from https://data.unep-wcmc.org/datasets/45 in May 2021). The final wetland area used in pyCIAM consists of the spatial union of these two datasets.

### 2.5.5 Sea Level Rise

We integrate local SLR projections from 23 different future scenarios drawn from six different global and regional sea level change research efforts conducted in recent years. These are detailed in Table 2. We model and present results for the median projections for each of these 23 future SLR scenarios in this paper, although we also ran pyCIAM using the 17th and 83rd percentile SLR runs for all 23 scenarios. The broad range of scenarios covered in our analysis (from 0.25 to 2m of GMSL rise in 2100) cover the plausible set of SLR trajectories; however, it can also be useful to assess the variation in impacts across different quantiles within a single scenario to assess uncertainty in impacts conditional on one emissions scenario. Such within-scenario assessment is outside of the scope of this manuscript but is an appropriate use of pyCIAM. To address this, results for the 17th and 83rd percentile of each SLR scenario are available in the model output dataset available on Zenodo (Section 5). We also note that pyCIAM is also configurable to run a probabilistic large ensemble of SLR trajectories on a multi-core computing platform, an approach used in recent research efforts using pyCIAM (Climate Impact Lab (CIL), 2022).

Our modeled future SLR pathways include the seven principal projections underlying the future sea level change trajectories detailed in the Intergovernmental Panel on Climate Change's (IPCC) Sixth Assessment Report (AR6) (Fox-Kemper et al., 2021). The data for these projections were generated using the Framework for Assessing Changes To Sea Level (FACTS, Kopp et al. (2023)) and were obtained from the report's public data repository (https://doi.org/10.5281/zenodo.6382554) (Garner et al., 2022). These seven trajectories represent different combinations of future emissions and underlying physical processes that influence sea levels. These scenarios are partitioned into two groups: *low confidence* (n=2) and *medium confidence* (n=5), which refer to the relative level of confidence of the underlying physical processes reflected in each future scenario. *Medium confidence* projections are considered to be of higher likelihood but do not incorporate deeply uncertain physical processes, such as marine ice cliff instability, that could have large impacts on future sea levels, particularly in higher emission scenarios. These processes are represented in the *low confidence* AR6 projections and project higher end-of-century GMSL values compared to their *medium confidence* counterparts (Table 2).

It should be noted that each of the AR6 emissions scenarios were originally constructed using Integrated Assessment Models (IAMs) driven by a single socioeconomic trajectory (i.e. a single SSP). However, when assessing economic impacts of climate change it is often useful to separate future changes in welfare caused by non-climate-related socioeconomic trends from climate impacts. This is done by holding baseline growth rates fixed across emissions scenarios. For this reason, we assess damages from each of the AR6 emissions scenarios under each of the 5 SSPs, even though some emissions trajectories may be more or less plausible under different SSPs.

We also incorporate the five main SLR scenarios represented in the U.S. interagency Sea Level Rise Technical Report (2022), led by the National Oceanic and Atmospheric Administration (NOAA) (Sweet et al., 2022) and derived from the FACTS-based projections in Garner et al. (2022). The SLR pathways in this report were organized by their projected GMSL value in 2100, rather than by global emissions trajectories. As such, they are grouped into five bins, based on different plausible GMSL values

in 2100: Low (0.3m), Intermediate-Low (0.5m), Intermediate (1.0m), Intermediate-High (1.5m) and High (2.0m)[6]. These data were obtained from the report's public data repository (https://doi.org/10.5281/zenodo.6382554).

The remaining 11 SLR projections are derived from the LocalizeSL framework (Kopp et al., 2014, 2017) (https://doi.org/10.5281/zenodo.6029807). LocalizeSL was used in the IPCC AR5 report (Church et al., 2013) and in subsequent publications (e.g. Kopp et al. (2017), Sweet et al. (2017), Rasmussen et al. (2018), Bamber et al. (2019), DeConto et al. (2021), Tebaldi et al. (2021)) prior to the introduction of FACTS. Similar to the AR6 SLR projections derived from FACTS, these based on LocalizeSL reflect a distribution across emissions scenarios, as well as across the component models used to represent the various contributing factors to SLR. These differences in component models refer to alternate assumptions and process representations regarding all contributors to sea level rise, with particularly influential differences in assumptions relating to ice sheet contributions.

Overall, these 23 scenarios cover a likely range of plausible SLR trajectories in the 21st century and allow us to estimate the marginal welfare costs of additional SLR across this full range (Fig. 4). Scenarios based on emissions trajectories may be most relevant for users interested in evaluating the benefits of emissions mitigation while those based on GMSL levels may be most relevant for local planners seeking to design adaptation strategies.

Notably, each of these scenarios contains a Monte Carlo sampling of a distribution of local SLR projections. However, because the 23 scenarios we reflect in this analysis cover a broad range of outcomes, for the purposes of this paper, we present results only for the median SLR projection at each coastal segment. In other words, the presented results reflect impacts in a world in which all regions experience the median projected RSLR for that scenario. Given the computational improvements in pyCIAM and its scalable design, it is suited for execution on a full Monte Carlo distribution. Climate Impact Lab (CIL) (2022), for example, applies CIAM to a 110,000-sample ensemble, using 10,000 draws from each of the 11 LocalizeSL-based SLR projections.

In addition to projections of climate change-induced SLR, and in alignment with Diaz (2016), we run a "no climate change" counterfactual scenario in which all SLR components are set to 0 except for a spatially heterogeneous and empirically estimated background rate of change parameter that includes drivers assumed to be unaffected by climate change (e.g. glacial isostatic adjustment, tectonics, sediment compaction, and other processes contributing to vertical land motion). This is a probabilistic parameter in the LocalizeSL and FACTS frameworks that is held fixed across all scenarios from a given modeling framework. The impacts estimated under these scenarios are subtracted from those in the climate change-driven scenarios to isolate the contributions of climate change to global 21st century coastal economic impacts (see Fig. 4).

To estimate local sea level extremes, we linearly combine the fixed ESL distributions from CoDEC with an annually interpolated version of the decadal SLR projections from each of these 23 scenarios. This allows us to maintain a globally consistent representation of extremes at reasonably fine resolution. Limitations of this "local bathtub" approach are described in Section 3.3.

---

[6]These GMSL values are expressed relative to GMSL in 2000, while pyCIAM expresses GMSL relative to 2005, making its end-of-century values associated with these scenarios in pyCIAM approximately 2cm lower (Table 2) than those specified in (Sweet et al., 2022).

| ID | SLR Scenario | Model Used | GMSL in 2100 [m] (median) |
|---|---|---|---|
| NCC | No Climate Change* | CIAM, pyCIAM | 0.00 |
| AR6-Med | IPCC AR6 *Medium Confidence* (2021): SSP1-1.9, SSP1-2.6, SSP2-4.5, SSP3-7.0, SSP5-8.5 | pyCIAM | 0.38, 0.44, 0.56, 0.68, 0.77 |
| AR6-Low | IPCC AR6 *Low Confidence* (2021): SSP1-2.6, SSP5-8.5 | pyCIAM | 0.45, 0.88 |
| Sweet | US Interagency SLR Technical Report (2022): Low, Int-Low, Int, Int-High, High | pyCIAM | 0.28, 0.48, 0.98, 1.48, 1.98 |
| K14 | Kopp et al. (2014): RCP 2.6, RCP 4.5, RCP 8.5 | CIAM, pyCIAM | 0.48, 0.58, 0.78 |
| SR | IPCC-SROCC (2019): RCP 2.6, RCP 4.5, RCP 8.5 | pyCIAM | 0.49, 0.60, 0.88 |
| B19 | Bamber et al. (2019): Low (2° C), High (5° C) | pyCIAM | 0.68, 1.10 |
| D21 | DeConto et al. (2021): RCP 2.6, RCP 4.5, RCP 8.5 | pyCIAM | 0.52, 0.62, 1.10 |

*Includes local background rates of relative sea level rise at each segment due to non-climatic background processes. Because of model differences, the FACTS-based projections (AR6 and Sweet) will use slightly different no-climate-change scenarios than those based on LocalizeSL.

**Table 2.** GMSL rise between 2005 and 2100 for each median SLR scenario used in the pyCIAM and Diaz (2016) models, representing the x-axis positions of costs by scenario displayed in Fig. 4.

Values for median GMSL rise throughout the 21st century are detailed in Table 2 below. For reference, an equivalent table for the 17th and 83rd percentile SLR projections for each scenario is provided as Supplemental Table C1.

## 2.6 Socioeconomic Variables

### 2.6.1 Population

In SLIIDERS, we use information from LandScan 2021 (Sims et al., 2022) to represent the present-day spatial distribution
of population. We then maintain this within-country distribution and scale the country totals to match the SSPs (Riahi et al., 2017), exponentially interpolated between 5-year projections to annual values. Because the SSPs begin in 2010 and pyCIAM begins in 2005, we must scale populations back to 2005. To do so, we use observed country-level growth rates from 2005 to 2010 to backcast from the 2010 SSP projections, which are constant across all SSPs. Observed rates are drawn primarily from the Penn World Table (PWT) 10.0 dataset (Feenstra et al., 2015), with missing countries filled through a variety of sources

including the 2022 UN World Population Prospects (United Nations, Department of Economic and Social Affairs, Population Division, 2022), multiple iterations of the CIA World Factbook (Agency, 2021), World Bank World Development Indicators (WDI, Bank (2021)), and local government statistics for some small island states. To project population forward For countries and territories not covered by the SSP data, we use global average population growth rates applied to 2010 estimates.

### 2.6.2 GDP

pyCIAM combines SSP-consistent, country-level GDP projections from two growth models - one from IIASA (Crespo Cuaresma, 2017) and one from the Organisation for Economic Co-operation and Development (OECD, Dellink et al. (2017)) OECD) and population projections from IIASA (Kc and Lutz, 2017) to create country-level GDP per capita projections. These data are available on the SSP Database (Riahi et al., 2017). SSP interpolation and extrapolation approaches match those used for population values. Observed values for 2005-2010 are again drawn from PWT 10.0 where available, with alternative sources

including (Fariss et al., 2022), OECD Regional Statistics (for Economic Cooperation and Development, 2020), the 2021 International Monetary Foundation World Economic Outlook (IMF, 2021), and the WDI. Where and when country-level estimates are unavailable but estimates do exist for associated sovereign entities, we use a regression estimator described in Bertram (2004) to estimate per capita GDP for the territories. For the countries and territories not covered by IIASA and OECD projections, we take the global-average per capita estimates in 2010 and interpolate/extrapolate using the global average yearly

growth rates for missing years.

    To create per capita GDP estimates ($ypc$) for coastal segments in pyCIAM for each year ($t$), we use the same national-to-segment downscaling approach as Diaz (2016), which relates population density to income. See Equation 8 in the Diaz (2016) Supplemental Information for further details. In Diaz (2016) population density is assumed to be homogeneous within segment, which implies that all elevation slices within a coastal segment are prescribed the same local income. In pyCIAM,

each elevation slice within each region has a unique population density. Thus, we apply this downscaling approach separately to each elevation slice.

### 2.6.3 Physical Capital

In addition to assessing the exposure of human population to SLR-related hazards, pyCIAM also assesses the exposure of physical capital stock to these threats. Both the IIASA and OECD GDP growth models utilize projections of physical capital;

however, neither model has publicly released these projections. Therefore, to create future capital stock estimates, we extract the relevant growth equations for OECD's Env-Growth model as described in Dellink et al. (2017). The capital growth trajectory in the IIASA model is exogenously specified, constant across SSP scenario, and yielded implausibly large capital stocks in later years. For instance, the IIASA model projects that Macau reaches $30.8 quadrillion in 2100 capital stock, which is 23 times that of the U.S. in 2100 ($1.3 quadrillion) and 200,000 times that of Macau in 2010 ($134 billion) (all values in constant

2019 PPP USD). Due to such implausible growth rates, we do not use the IIASA capital growth trajectory in pyCIAM.

    We use country-level capital stock estimates up through 2020 and then use 2020 estimates as the initial conditions for this growth model. Like with population and GDP, historical estimates of capital come primarily from PWT 10.0. Where these

values are missing and outside of the special cases of Cuba and North Korea, SLIIDERS uses estimates of the ratios of non-financial wealth (NFW) to GDP derived from the 2022 Credit Suisse Global Wealth Databook (Credit Suisse Research Institute, 2022), combined with nominal GDP information from United Nations System of National Accounts (UNSD, 2021). Following the approach taken in (Eberenz et al., 2020), we then multiply PPP GDP by these NFW-to-GDP ratios to acquire proxies of physical capital. For Cuba, we use the ratio of Cuban and the U.S. capital stock values from Berlemann and Wesselhöft (2017) and multiply this ratio with the U.S. capital stock values from PWT 10.0. For North Korea, we multiply the capital-to-GDP ratio in Pyo and Kim (2020) with PPP GDP.

Then, we apply the OECD capital stock equations with the estimated 2020 capital stock values and SSP-consistent GDP projections to obtain projections of capital stock for each SSP scenario and for each GDP growth model. To parameterize these equations, we use a value for the partial elasticity of GDP with respect to capital taken from Crespo Cuaresma (2017) (0.326), since this is not reported in Dellink et al. (2017). We also estimate country-specific initial conditions for the marginal product of capital using a modified Cobb-Douglas production function fit to the historical capital data. See Sect. A3 for further methodological detail.

pyCIAM uses the LitPop dataset (Eberenz et al., 2020) to represent within-country spatial distribution of physical capital stock at 30 arc-second resolution. LitPop combines population information from the Gridded Population of the World dataset (v4.1) (University, 2016) with nightlight intensity (Román et al., 2018) to downscale country-level estimates of total physical assets. In some countries, e.g. Libya and Syria, LitPop does not provide any downscaled estimates. In these locations, we use the downscaled estimates provided by the GEG-15 dataset (Bono and Chatenoux, 2014). For the small number of island countries that do not have capital distributions reflected in either dataset, we assume homogeneous capital stock.

In pyCIAM, the ratio of mobile to immobile capital is used to determine costs of inundation. Diaz (2016) used a fixed ratio of 10%. However, PWT 10.0 contains country-level information that can be used to estimate across-country heterogeneity in this ratio. PWT decomposes physical capital into four categories:

1. Residential and non-residential structures
2. Machinery and non-transport equipment
3. Transport equipment
4. Other assets

For SLIIDERS, we assume that the first category (residential and non-residential structures) represents immobile capital and the others represent mobile capital. We take the average mobile fraction from 2000-2019 and apply this at the country level. These country specific values vary from 1% (Haiti) to 52% (Equitorial Guinea) with 25th, 50th, and 75th percentiles of 14, 18, and 20%, respectively.

### 2.6.4 Construction Costs

We maintained the same reference unit cost of coastal protection utilized in CIAM but updated the national construction cost index scaling factors by using the ratio of construction cost indices from ICP 2017 (World Bank, 2020) instead of 2011. For

countries not included in this dataset, we augment with the country-level construction cost indices used in Lincke and Hinkel (2021), averaged across the rural and urban distinction.

## 2.7 Other Features

### 2.7.1 Model Duration

Diaz (2016) runs from 2000-2200. However, the SSPs stop at 2100 and thus the SLIIDERS dataset does as well. Because of this, and because the AR6 SLR scenarios begin in 2005, we limit pyCIAM to 2005-2100. Using the 4% discount rate employed in Diaz (2016) and pyCIAM, the discount factors for 2100-2200 costs vary from 2% in 2100 to 0.03% in 2200, so the exclusion of these additional years is unlikely to have a substantial effect on the optimal adaptation option selected by each segment.

### 2.7.2 Timesteps and Planning Periods

We increase temporal resolution from the decadal timesteps used in Diaz (2016) to annual. In addition to the exponential interpolation of 5-year SSP inputs described above, decadal SLR projections are linearly interpolated to yield annual values. The 40-50 year planning periods used in Diaz (2016) yield substantial step-changes in realized costs at mid-century and end-of-century due to substantial simultaneous global adaptation actions. To generate a smoother time series of costs, we use decadal planning periods. A potential trade-off of using shorter planning periods is that this may overestimate the frequency with which 615 governments and populations are able to update major adaptation actions. An unrealistically agile representation of large-scale adaptation actions may underestimate associated net present cost because some adaptation costs can be postponed to future years with lower discount factors. Future work may empirically estimate the frequency at which adaptation approaches are updated and explore further options for incorporating planning periods that are not globally simultaneous and thus do not lead to substantial step-changes in global SLR costs at the start of each period.

### 2.7.3 Net Present Value Calculation

In Diaz (2016), the NPV each segment uses to calculate an optimal adaptation approach is calculated from 2010-2200, excluding the initial planning period of 2000-2009. In this way, each segment is allowed a "free" initial relocation or protection action. For example, if a segment chooses to protect to the 1-in-10,000 year sea level height, which is 3 meters in 2000, they do not consider the costs of building a corresponding seawall when calculating the NPV of this action. They only consider the 625 marginal cost of extensions to this seawall to remain at the 1-in-10,000 year height as local sea levels increase.

The rationale for this initial "spin-up" period in Diaz (2016) was to allow each segment to choose an optimal adaptation approach without including costs for adaptation measures that may already exist but are not reflected in observed values due to the lack of high quality global input data describing population distribution and coastal protection measures. In other words, segments were allowed to choose their optimal adaptation approach based only on adaptation costs associated with *updating* 630 adaptation (e.g. through height increases of protection or additional managed retreat) but not based on the costs of initial implementation (e.g. the initial protection construction or managed retreat).

By using finer resolution population and capital stock estimates, SLIIDERS partially ameliorates this need by providing more accurate observed measures of coastal exposure. In addition, we argue that any existing adaptation measures would have to have been implemented at some point in history when they were presumably determined to be a cost-effective approach, even including the initial costs of implementation. This deviates from the assumption in Diaz (2016) that such initial adaptation does not incur costs, which we believe is likely to overestimate the state of present-day adaptation. Including the costs in this "spin-up" period when calculating NPV, along with calibrating the non-market costs of relocation (see Section 2.3), reduces the amount of instantaneous relocation observed under the optimal adaptation scenario.

For these reasons, in the configuration of pyCIAM presented here, each segment uses costs from the entire model duration of 2005-2100, inclusive of the initial adaptation costs, to calculate NPV and choose an optimal adaptation approach. This configuration is applied to costs from all scenarios, including those from the "no climate change" counterfactual scenario that are subtracted from the "with climate change" scenarios in order to isolate the climate change contributions to coastal welfare impacts.

Because of this consistency in application, the choice of the initial NPV year is likely to have a minimal effect on the estimated climate change costs. However, it will substantially affect the "un-differenced", total costs associated with both the "with climate change" and "no climate change" in the initial adaptation period. This is reflected in substantially different NPV calculations between this paper and Diaz (2016) in this un-differenced context (see Fig. B1). pyCIAM provides users with a configurable parameter to determine whether initial adaptation costs should be accounted for in each segments NPV calculation or not.

In addition to modifying the starting year of the NPV calculation, we make one change to the application of a discount rate. Diaz (2016) applied the discount rate at the start of each decadal timestep to the full 10 years of costs incurred in that timestep. This approximation overestimates the discounted cost for all years after the first. We avoid this issue by using annual timesteps; however, when comparing NPV results to Diaz (2016) (Fig. 4), we apply annually varying discount rates to the Diaz (2016) outputs as well.

### 2.7.4 Manual Correction Factors

In pyCIAM, the following manual correction factors in the original code underlying Diaz (2016) have been removed. These correction factors were originally used by Diaz (2016) in order to correct for certain limitations in data availability or quality that are no longer necessary after incorporating the data updates in SLIIDERS:

1. Doubling the price of construction on all "island" segments. The new construction cost index values utilized in pyCIAM should reflect any increased construction costs on island nations. Additionally, segments defined as "island" in CIAM were not entirely consistent, with some islands receiving the label and others not.

2. Halving the protection heights under the protection adaptation scenario corresponding to 10-year ESL heights. This was originally implemented to account for elevation profiles found in the GLOBE DEM that were deemed physically implausible (extremely high area totals from 0-1m), but is no longer required following the updated CoastalDEM elevation values.

3. Averaging of the inundated land area-by-elevation bins for the first two (0-1m, 1-2m) bins in order to smooth the elevation profile due to the high 0-1m area totals in the GLOBE DEM values. This adjustment, too, is no longer required following the updated CoastalDEM elevation values.

## 3  Results and Discussion

Upon implementing the changes described above, global costs estimated by pyCIAM diverge modestly from those in Diaz (2016). Additionally, we obtain estimates for a greater breadth of socioeconomic and SLR trajectories that reflect deep uncertainty in these processes. Fig. 4 displays estimated global costs for the following global SLR-driven cost metrics reported in Diaz (2016): (i) global net present costs under an optimal adaptation scenario using a 4% discount rate, (ii) end-of-century annual total costs under that same scenario, (iii) end-of-century annual total costs under a "reactive retreat only" scenario, and (iv) end-of-century annual costs of wetland loss under the optimal adaptation scenario. Global NPV and end-of-century costs for the highlighted scenarios in Fig. 4 and for a "middle of the road" socioeconomic growth scenario (SSP2/IIASA) are shown in Table 3.

Results are shown for the pyCIAM model both in its replicated CIAM configuration and after all the above changes were applied. Values are expressed such that each vertical group of points comprise the spread of results between the different socioeconomic projections for a given SLR scenario, with the position along the x-axis representing that scenario's median GMSL value in 2100. As described in Sect. 2.5.5, all of the pyCIAM results use a constructed "median" SLR trajectory where each location experiences the median RSLR across the probabilistic projected distribution. This matches the approach used in Diaz (2016).

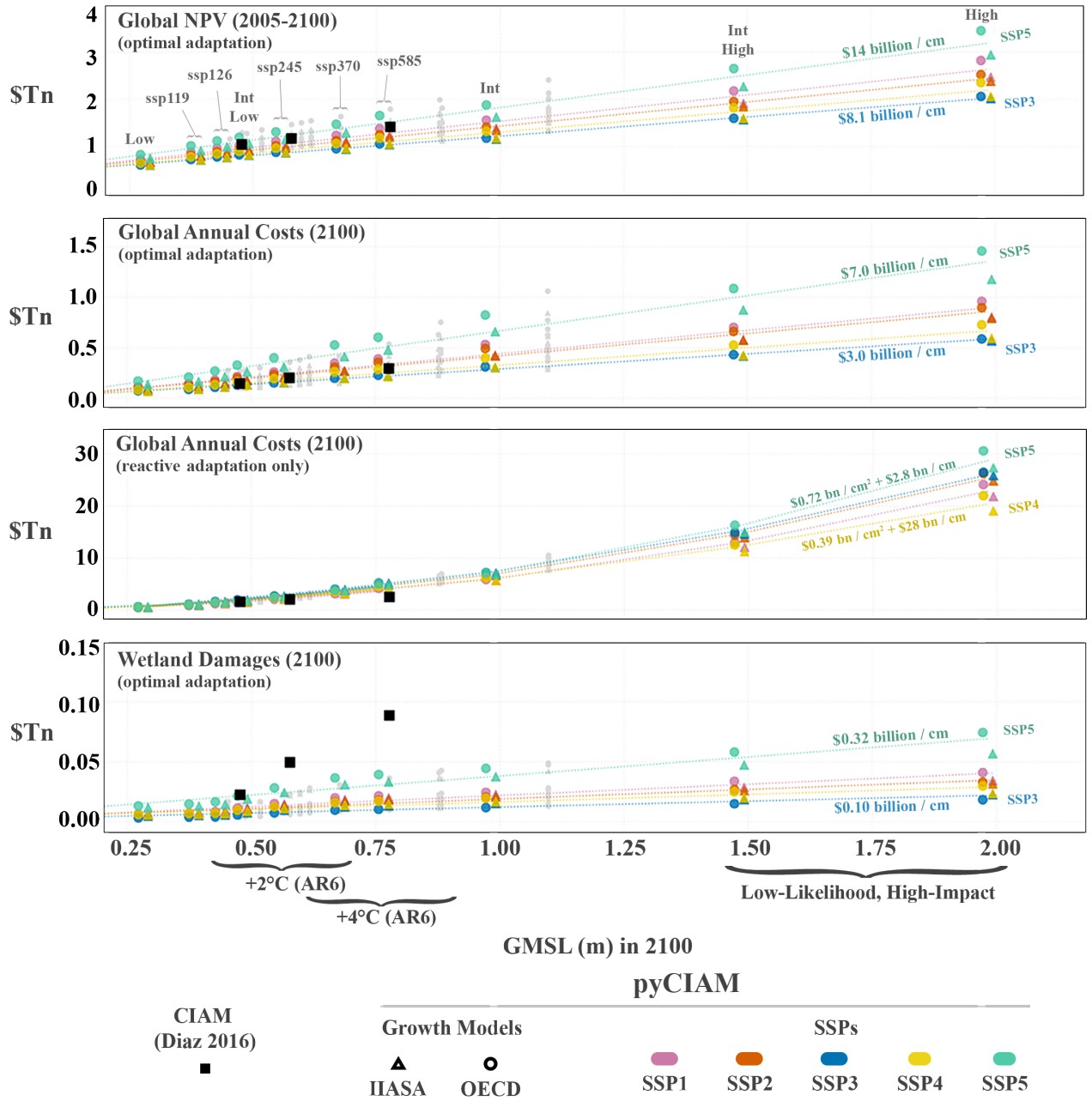

**Figure 4.** Comparison of global cost estimates under each SLR scenario. Values are costs from climate change induced SLR only, i.e. after differencing the costs under a "no climate change" scenario that reflects median projections of non-climatic RSLR rates and no GMSL rise. All costs are expressed in constant 2019 PPP USD. Each vertical group of points describes a single SLR scenario, with each point in the group representing a unique combination of SSP and economic growth model. For visual clarity, only *medium confidence* AR6 and Sweet et al. (2022) scenarios are indicated with colored markers and jittered slightly along the x-axis based on runs using the OECD (-1cm) or IIASA (+1cm) economic growth model. The remaining SLR scenarios are shown in grey without jitter. Dashed lines represent fitted relationships between the cost metric and 2100 GMSL across the full set of SLR scenarios. Relationships are estimated for each SSP scenario and are linear for all metrics except for global annual costs under a reactive adaptation scenario.

| SLR Scenario | GMSL [m] (2100) | NPV $Tn (bp) *Optimal* | NPV $Tn (bp) *Reactive* | Costs (2100) $Tn (bp) *Optimal* | Costs (2100) $Tn (bp) *Reactive* |
|---|---|---|---|---|---|
| Low (Sweet et al.) | 0.28 | 0.66 (1) | 1.21 (3) | 0.09 (2) | 0.56 (11) |
| SSP1-1.9 (AR6-Med) | 0.38 | 0.80 (2) | 2.00 (4) | 0.12 (2) | 1.03 (20) |
| SSP1-2.6 (AR6-Med) | 0.44 | 0.86 (2) | 2.41 (5) | 0.15 (3) | 1.47 (28) |
| Int-Low (Sweet et al.) | 0.48 | 0.91 (2) | 2.46 (5) | 0.18 (3) | 1.74 (33) |
| SSP2-4.5 (AR6-Med) | 0.56 | 0.98 (2) | 3.28 (7) | 0.21 (4) | 2.46 (47) |
| SSP3-7.0 (AR6-Med) | 0.68 | 1.08 (2) | 4.15 (9) | 0.27 (5) | 3.66 (69) |
| SSP5-8.5 (AR6-Med) | 0.77 | 1.20 (3) | 5.24 (11) | 0.31 (6) | 4.79 (91) |
| Int (Sweet et al.) | 0.98 | 1.34 (3) | 6.19 (14) | 0.42 (8) | 6.64 (126) |
| Int-High (Sweet et al.) | 1.48 | 1.84 (4) | 13.62 (30) | 0.58 (11) | 13.93 (264) |
| High (Sweet et al.) | 1.98 | 2.38 (5) | 24.61 (54) | 0.79 (15) | 24.85 (471) |

**Table 3.** Global estimated NPV (2005-2100) and annual costs of climate-driven SLR in 2100, expressed in constant 2019 PPP USD, for the *medium confidence* AR6 and Sweet et al. (2022) SLR scenarios. Each metric is presented for both the optimal adaptation and reactive retreat modeling configurations, using the SSP2/IIASA socioeconomic growth model. Numbers in parentheses show the fraction of global GDP associated with these costs under the SSP2/IIASA growth scenario in units of basis points (1/100ths of a percent). For columns 3 and 4, the NPV of GDP 2005-2100 is used for this calculation; for columns 5 and 6, GDP in 2100 is used.

## 3.1 Total SLR Costs

The global distribution of end-of-century average annual costs of climate-driven SLR under optimal adaptation, aggregated to first-level administrative regions (equivalent to state-level in the U.S.), is shown in Fig. 5, using the AR6 (*medium confidence*) SSP2-4.5 SLR scenario and SSP2-IIASA socioeconomic trajectory. Fig. 6 similarly demonstrates spatial heterogeneity in the annual cost savings realized through optimal adaptation, relative to costs in the reactive retreat scenario.

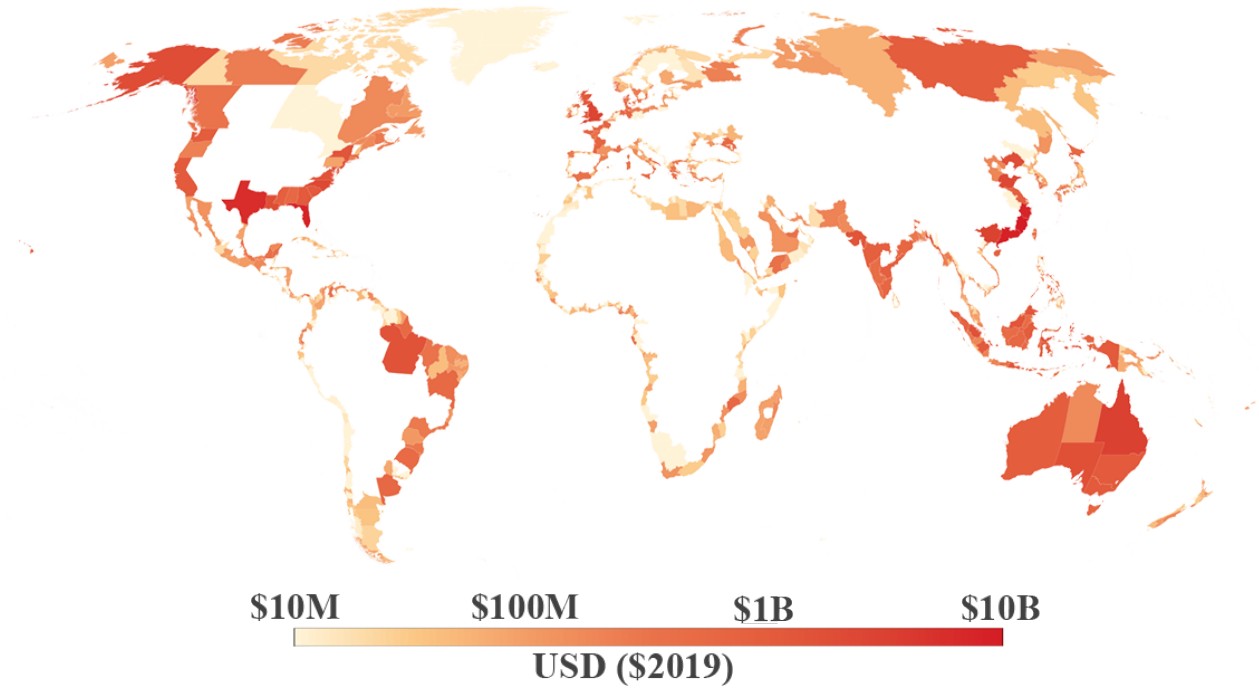

**Figure 5.** Example estimated annual average costs in 2100 by first-level administrative region (equivalent to state-level in the U.S.). Results shown reflect optimal adaptation, using the AR6 (*medium confidence*) SSP2-4.5 SLR scenario and SSP2, IIASA socioeconomic projections.

Median global NPV values from 2005-2100 under optimal adaptation ranges from $600 billion to $3.4 trillion in pyCIAM across its 230 SLR-SSP-economic growth model scenarios, corresponding to end-of-century GMSL rise values between 0.28 and 1.98m, relative to 2005 mean sea level (Fig. 4). Estimates of global NPV from Diaz (2016) range from $1.0 to $1.4 trillion in the three SLR scenarios considered (end-of-century GMSL rise from 0.48 to 0.78m, Table 2). Comparing the three SLR scenarios used in pyCIAM that match those employed in Diaz (2016) (K14 RCPs 2.6, 4.5, 8.5), pyCIAM's median global NPV values from 2005-2100 are similar to those estimated by CIAM, with some socioeconomic projections yielding higher estimates and some yielding lower (Fig. 4, Table C2).

When considering total damages, rather than the difference between them, pyCIAM estimates significantly higher global NPV (∼5-6x) and moderately higher end-of-century costs (∼2x) compared to Diaz (2016) (Fig. B1). There are several reasons for these differences. First, the decision to include initial adaptation costs in the NPV calculation and optimal adaptation selection for each segment contributes to the substantially higher NPV values seen in pyCIAM (see Section 2.7.3). Second, we use a calibrated value for non-market relocation costs almost an order of magnitude larger than that used in Diaz (2016) (see Section 2.3). This drives more segments toward choosing protection and thus drives up global construction and maintenance costs in addition to relocation costs. Third, Diaz (2016) assumes that all abandoned capital has fully depreciated by the time of abandonment for proactive retreat scenarios, while pyCIAM avoids this assumption due to a lack of empirical evidence

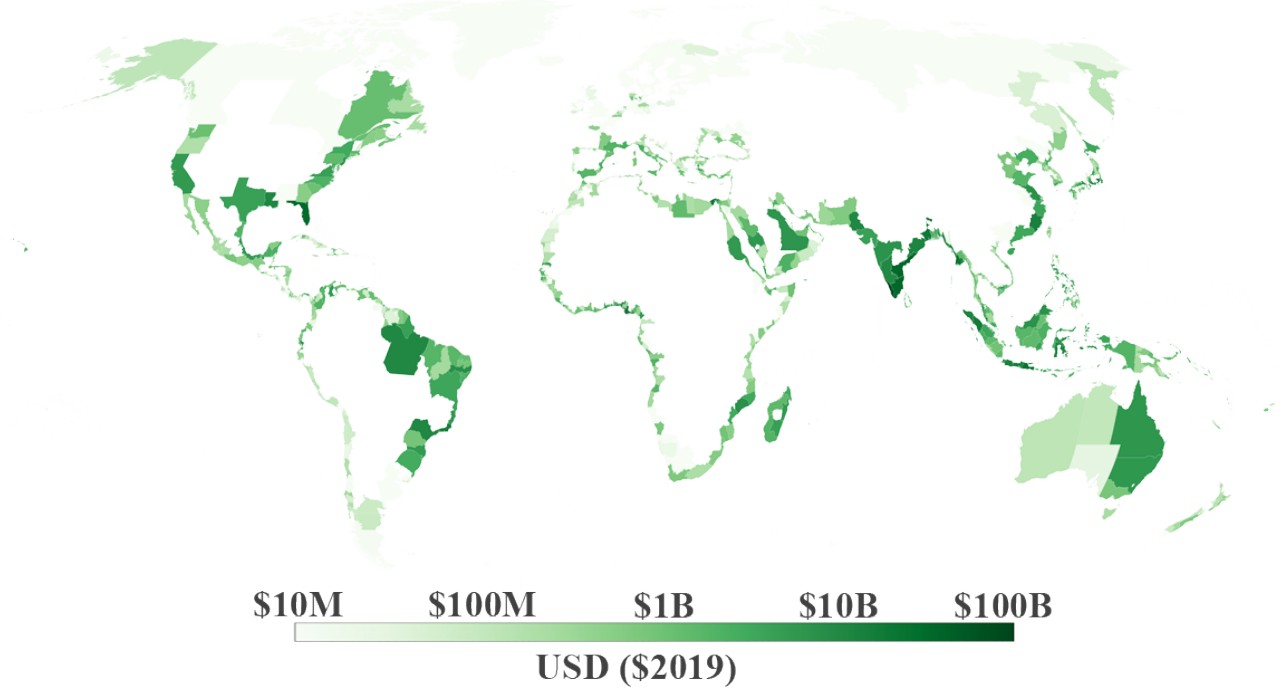

**Figure 6.** Example estimated annual adaptation benefits in 2100 by first-level administrative region (equivalent to state-level in the U.S.). Results shown reflect the AR6 (*medium confidence*) SSP2-4.5 SLR scenario and SSP2, IIASA socioeconomic projections.

(Section 2.2.1). Fourth, in Diaz (2016), segments choosing reactive adaptation were assumed to retreat at least up to a height deemed optimal under current sea levels. This often led to retreat higher than mean sea level in order to minimize ESL-related damages; however, land abandonment and relocation costs were not assessed for this full retreat height. Instead, they were only assessed up to mean sea level, lowering estimated costs for these two cost types. Fifth, Diaz (2016) reduced the 10-year protection height by 50% for all segments as an ad hoc adjustment to account for an implausibly large land area contained in the 0-1m elevation slice, as reported by DIVA and derived from GLOBE DEM (Section 2.7.4. Sixth, projected capital stock and population in SLIIDERS across its SSP and growth model scenarios are significantly higher than those modeled in Diaz (2016). For example, the mid-century global capital stock located between 0 and 15 meters above sea level ranges from $220 to $370 trillion (2019 USD) across the five SSPs and two growth models in SLIIDERS, compared to $97 trillion in Diaz (2016). Similarly, SLIIDERS' mid-century population ranges from 1.19 to 1.35 billion people across the five SSPs (population is equivalent in each economic growth model), compared to 1.18 billion in Diaz (2016). The SSP-based ranges differ most from the Diaz (2016) trajectories around mid-century before beginning to converge toward end-of-century. Finally, higher modeled costs in pyCIAM may also be driven by updated topographic maps and other physical input datasets used for estimating exposure to SLR in pyCIAM.

Annual global costs due to climate-driven SLR in 2100 under optimal adaptation range from $70 billion to $1.5 trillion across all pyCIAM scenarios, and from $100 billion to $540 billion across the K14-pyCIAM scenarios that correspond to those used in Diaz (2016). The corresponding values from Diaz (2016) range from $150 billion to $290 billion, with the smaller range being largely driven by Diaz (2016) considering only one socioeconomic growth scenario. Under a reactive retreat scenario, pyCIAM values are generally similar to those of Diaz (2016) for low and medium-SLR scenarios and higher for high-end SLR scenarios (Fig. 4).

Considering the AR6 (*low* and *medium-confidence*) warming scenarios and associated ranges of global SLR by 2100, we estimate that under $2°C$ of warming by 2100 (+0.40-0.69m GMSL), annual end-of-century costs will be between $110 billion and $530 billion (between 0.02 and 0.07% of global GDP), depending on SSP, economic growth model, and SLR magnitude and assuming optimal adaptation. For AR6's $4°C$ scenario (+0.58-0.91m GMSL), these costs range from $200 billion to $750 billion (0.04 to 0.09%). Also, for two low-likelihood, high-impact scenarios (Sweet-IntHigh, Sweet-High), which incorporate more uncertain physical processes like accelerated marine ice sheet and marine ice cliff instability, and correspond to GMSL rises of 1.5-2.0m by 2100, global annual costs range from $420 billion to $1.5 trillion (0.08 to 0.20%) by end-of-century under the same set of assumptions.

Upon projecting costs across this wide range of SLR scenarios, we find a strongly linear relationship for both NPV and annual end-of-century wetland and total damages with respect to end-of-century GMSL. Depending on socioeconomic projections, the marginal NPV costs associated with 1 cm of end-of-century GMSL range from $8 billion to $14 billion, the marginal annual end-of-century total costs range from $3 billion to $7 billion, and the marginal annual end-of-century wetland costs range from $110 million to $350 million. In a scenario with only reactive adaptation, annual end-of-century costs are not only much higher in absolute terms but also increase in a much sharper (quadratic) manner with respect to GMSL.

## 3.2 Adaptation Costs and Benefits

The results of this analysis support the finding of Diaz (2016) that adaptive measures (through protection or retreat) can dramatically reduce the cost of sea level rise. For a GMSL rise of one meter by 2100 and a "middle-of-the-road" socioeconomic growth trajectory (SSP2/IIASA), optimal adaptation would reduce the NPV of coastal impacts by about $5 trillion, inclusive of these adaptation costs. This represents $\tilde{0}.9\%$ of the net present value of GDP over that same time horizon. Similarly, it would reduce average annual costs at end-of-century, inclusive of adaptation costs, by $\tilde{\$}6$ trillion (1.2% of end-of-century annual GDP) 3. This would require substantial global investment in protection ($770 billion NPV under the same scenario) and retreat ($310 billion NPV, including both market and non-market costs of relocation). Across all socioeconomic and SLR scenarios modeled, we find that optimal adaptation can lower the NPV of impacts by a factor of 1.6 to 12, relative to a reactive adaptation approach.

The global distribution of optimal adaptation strategies is displayed in Fig. 7 for the AR6 (*medium confidence*) SSP2-4.5 SLR and SSP2, IIASA scenario. Notably, the majority of segments that protect are located in Asia, where coastal population densities are generally high and construction costs, at least as parameterized by CIAM and pyCIAM, are relatively low. Scattered high-density areas across OECD countries in Europe and North America are protected as well. The fact that most protecting segments

opt for the maximum level of protection (1-in-10,000-year ESL height) also suggests that, for segments where protection is optimal, the parameterized marginal costs of building higher protection are almost always lower than the benefits they provide, up to the point where the protection heights have provided safety from an exceedingly rare event. Future work should develop

approaches to empirically calibrate the construction cost functions used in Diaz (2016) and ported to pyCIAM, as these may control the spatial distribution of protection. Similar to the pattern of maximizing protection, there is a common preference to retreat to the 1-in-10-year ESL height amongst segments that adopt retreat as their optimal strategy. This suggests that increasing the resolution of retreat options around this level may better reflect heterogeneity in optimal retreat height. Finally, segments for which reactive retreat is optimal are generally sparsely populated or unpopulated.

Fig. 8 displays the proportion of global segment populations adopting different adaptation strategies (protection, proactive retreat, and reactive retreat), across the various socioeconomic and SLR scenarios for both pyCIAM/SLIIDERS and CIAM. In general, while CIAM indicates that roughly 50% of the world's population would be protected under optimal adaptation and 50% would be relocated, pyCIAM, paired with SLIIDERS inputs, finds these ratios to be closer to 80% and 20%, respectively. This is largely due to our increased relocation cost parameter (Sect. 2.3), which disincentivizes retreat relative to protection. In

contrast to the influence of relocation cost on adaptation type, little variation is observed in these percentages across pyCIAM's different socioeconomic scenarios (Fig. 8). This stability is visible even within individual segments' adaptation choices and suggests that particular choices of adaptation strategy (protection versus retreat) and the return value to which the chosen adaptation strategy is enacted may be robust to a range of future socioeconomic and SLR trajectories for most coastal regions. Similar results are shown normalized by coastline length rather than population in Fig. B2.

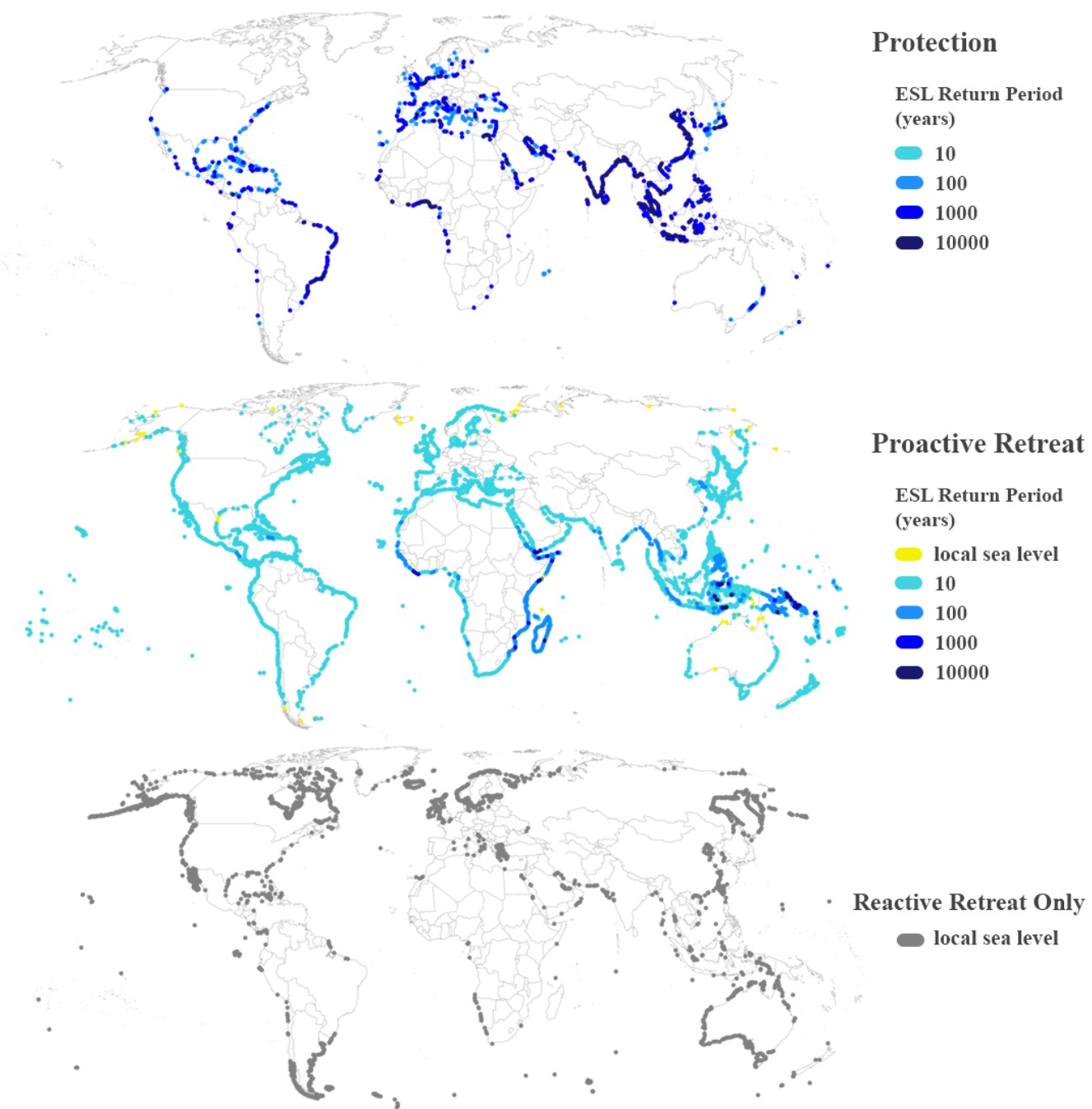

**Figure 7.** Adaptation strategies chosen by each segment in the optimal adaptation scenario. Each segment is represented by a marker at its centroid. Results reflect the AR6 (*medium confidence*) SSP2-4.5 SLR scenario and SSP2 and IIASA socioeconomic growth projections. Return periods indicate the level of protection/retreat that is adopted by each segment.

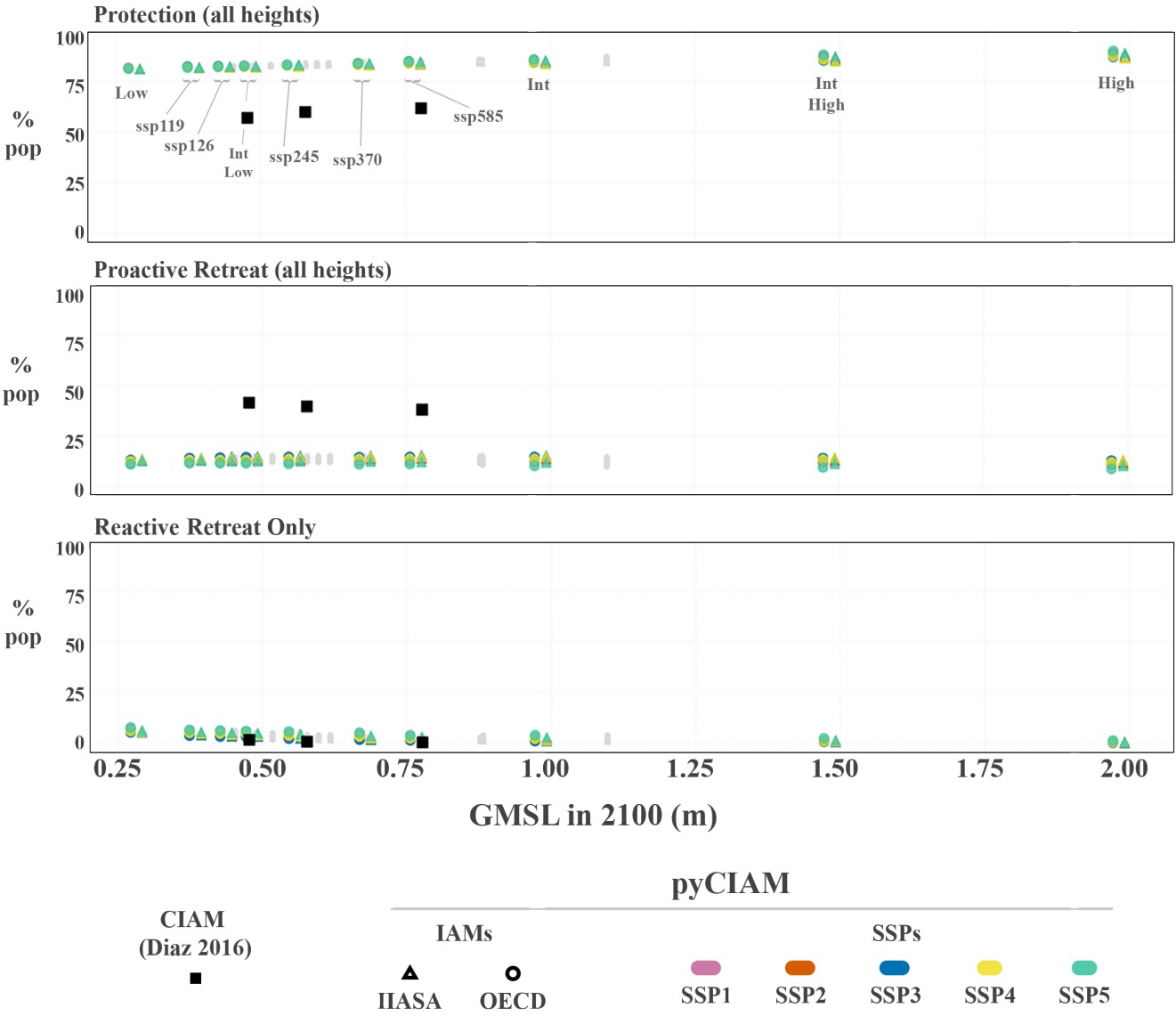

**Figure 8.** Comparison of optimal adaptation strategies adopted across all segments. Values represent percentages of global population residing at elevations below 15 meters in segments adopting each respective adaptation strategy. Proactive retreat and protection values are aggregates of all possible heights for each. Solid black squares represent the results from Diaz (2016). For visual clarity, only *medium confidence* AR6 and Sweet et al. (2022) scenarios are indicated with colored markers and jittered slightly along the x-axis based on runs using the OECD (-1cm) or IIASA (+1cm) economic growth model. The remaining SLR scenarios are shown in grey without jitter.

## 3.3 Model Limitations and Planned Improvements

pyCIAM is subject to some of the same limitations as its predecessor CIAM. First, adaptation is limited to the ten possible options introduced in Diaz (2016) — four protection heights, five proactive retreat heights and a reactive retreat action. Second,

segments are only allowed one protection or retreat standard throughout the model duration. They cannot, for example, retreat to the 1-in-10 year ESL height for the first 20 years and then retreat to the 1-in-100 year height. Rather a single optimal standard is chosen given the full distribution of potential future outcomes. Similarly, segments cannot combine both retreat and protection. More flexible approaches may enable lower-cost outcomes (Kopp et al., 2019; Haasnoot et al., 2019), though computational constraints have limited the implementation of more dynamic adaptation approaches to models with local domains (Lickley et al., 2014). A preliminary approach to this problem, such as allowing for a one time, mid-century alteration of adaptation strategies, could be a simple scheme to allow for some level of dynamic adaptation strategies. Third, insurance, subsidies or other policies may discourage proactive retreat even when the NPV would be positive, and these interventions are not taken into account by segment agents in the model when determining the least cost adaptation path. Fourth, many cost functions and parameters in the model are based on limited empirical evidence, as little evidence at fine resolution and global scale is available to inform the magnitude and heterogeneity of these costs.

Existing coastal protections are not directly modeled due to a lack of globally consistent data. Instead, existing protections are assigned in the model like those of any other year based on the least cost adaptation scenario for each segment. This means that protection costs in the initial year of the model will include the cost of constructing these existing structures, though these additional costs will be differenced out of our climate impact estimates because they will occur in both the "with climate change" and "no climate change" SLR trajectories.

Retreat or protection heights within each decadal planning period are chosen under perfect foresight of projected RSLR at that segment during the entire period, such that any maximum projected change in ESL return values due to RSLR is perfectly anticipated and incorporated into adaptation cost considerations and decisions. Notably, segments also chose their optimal adaptation strategy (e.g. protection to the 1-in-100 year ESL height) based on an NPV calculation that utilizes perfect foresight over the entire model duration. While this assumption cannot be correct in its extreme form, Fig. 8 suggests that these choices are very robust to uncertainty in future sea level and socioeconomic change.

pyCIAM also does not currently represent accommodation measures (e.g., infrastructure hardening and building elevation), which in some cases may be more cost-effective than either protection or retreat Oppenheimer et al. (2019); Kopp et al. (2019); Rasmussen et al. (2020). Accommodation encompasses a broad range of actions and is thus difficult to parameterize within the model. To our knowledge, accommodation is not represented in other coastal modeling platforms but could be the subject of future updates to pyCIAM. Additionally, the potential changing feasibility of both adaptation and accommodation measures in future decades, due to potential factors related or unrelated to climate change, like shifting supply chain and/or labor market dynamics, are not currently represented. These may prove to be relevant to society's capacity to effectively adapt in the future.

Our current estimation of the non-market costs of relocation detailed in Sect. 2.3 is intended to represent the fact that many coastal areas are observed to currently be under-adapted to present ESL hazards (Houser et al., 2015; McNamara and Keeler, 2013; McNamara et al., 2015; Armstrong et al., 2016; Haer et al., 2017; Hinkel et al., 2018; Suckall et al., 2018; Lorie et al., 2020). For example, Mendelsohn et al. (2020) estimated the cost-benefit ratio of building seawalls to be at least 2:1 in East Haven, CT, Council (2017) estimates this ratio for elevating coastal homes up to 9:1 in some U.S. locations, and Bakkensen and Mendelsohn (2016) found that the U.S., in particular, may be up to 14x less adapted to tropical cyclone hazards than other

OECD countries threats presently. Improved estimates of these non-market relocation costs could potentially be guided by more detailed empirical assessments of present-day under-adaptation to coastal hazards. While some of this under-adaptation is rationalized by our non-market costs of relocation, other factors including challenges of permitting and funding costly infrastructure projects, subsidized insurance (Craig, 2019) or limited risk information may play a role as well. We are aware of efforts to further understand adaptation costs and the reasons for under-adaptation (Bower and Weerasinghe, 2021; Berrang-Ford et al., 2021), but the current extent of empirical evidence quantifying sub-optimal adaptation is limited. If and when such evidence is available, the modularity of pyCIAM enables future integration of these estimates to improve its adaptation cost-benefit implementation.

Better global data describing existing coastal protection infrastructure would improve the accuracy of pyCIAM. Spatially resolved data on constructed protection around the globe is sparse. To overcome this, some studies assume a certain level of protection as being present in all coastal regions, making stylized assumptions based on population densities and national GDP (Sadoff et al., 2015). Other studies develop statistical models to empirically ground such relationships (Scussolini et al., 2016), and these have been incorporated in other global coastal adaptation models (Tiggeloven et al., 2020) and could be evaluated for use in future versions of pyCIAM. Further improvements to certain regions could also be made using protection data collected by Hallegatte et al. (2013) for 136 coastal cities.

Our reflection of local mean and extreme sea levels is limited by the resolution of our local MSL projections (1 degree in FACTS, 2 degrees in LocalizeSL) and our ESL distributions from CoDEC (50km coastline spacing). Because of the desire to build a globally consistent model using these inputs, we employ a "local bathtub" model in which all points nearest to a given pair of ESL and MSL prediction points receive the same mean and extreme sea level projections. While this local model preserves the substantial large-scale spatial heterogeneity in SLR and ESL, sub-grid-scale variation is ignored. In particular, bathtub models are known to overestimate storm surge in inland areas largely due to the deceleration of flows caused by surface roughness (Bootsma, 2022; Vousdoukas et al., 2016). A more sophisticated, dynamic representation of ESL based on local hydrodynamic simulations for each MSL/ESL combination is beyond the computational scope of this analysis but may yield improved future results and could be incorporated either "on-the-fly" within the pyCIAM model or in a pre-processing step that updates the ESL distributions in SLIIDERS.

Because pyCIAM linearly combines present-day ESL estimates and SLR predictions, our current approach also ignores changing ESL distributions due to (a) climate-driven changes to storm surge distributions from, for example, altered tropical cyclone frequency and intensity; and (b) the dynamic interaction between storm surge and MSL, moderated by local topography.

Despite these limitations in estimating sea levels, it is important to note that when isolating climate change-induced coastal costs, we difference the costs of a no-climate change baseline scenario that uses the same local bathtub flood model. This differencing also serves as a bias correction step, partially mitigating any over-estimates of flooding damages potentially introduced by the bathtub approach, though some high or low bias may still be present in the final results. Total (un-differenced) cost estimates (Fig. B1), however, will reflect any bias associated with the bathtub flood model. Accounting for these future changes is important for planning purposes, but represents a major computational challenge.

Additional geophysical dynamics associated with SLR inundation and related flooding, such as erosion, salinization of aquifers and estuaries, are also not currently addressed in our approach. Finer-scale wave setup and ESL behavior within

845 complex coastlines at the sub-segment scale could also be useful to capture in future modeling. This would require estimates of ESLs at a much higher spatial resolution than is provided in the CoDEC dataset and is therefore currently infeasible given available input data.

Future development that refines the spatial resolution of our coastal segments from the current 50 km spacing would enable a finer analysis of the local dynamics of hazard, exposure, and potential adaptation decisions for hyper-local decision making

entities. At present, such an effort is hampered by a lack of more granular global inputs used to generate the SLIIDERS dataset. In particular, the 50 km-resolution CoDEC ESL dataset and the one (FACTS) or two (LocalizeSL) degree resolution of the SLR projections represent the best available observations and projections of sea levels at global scale. Using finer-resolution coastal segments would provide limited gains in model precision as the only sub-50km variation would be from enabling decision-making to occur at finer scales, rather than the incorporation of higher-resolution hazard information. The resolution at which

protection and retreat decisions are and will be made likely varies substantially around the world, and thus it is unclear whether such an approach would yield more or less valid adaptation projections. As the quality and resolution of related input datasets evolve, SLIIDERS and pyCIAM can be updated to reflect these advances.

Finally, our hydraulic connectivity model masks only those regions that are not connected at 20 meters of SLR relative to 2005 levels. Some areas may not meet this criteria but still may be non-connected at lower sea levels. For example, a location

that is at 1 meter above sea level but is behind a hill at 2 meters above sea level would be flooded by our model for sea levels of 1.5 meters. Future work could address this by assigning each pixel not only an elevation but a barrier height that would be treated similar to how manmade protection heights are treated in pyCIAM. This would increase the dimensionality of several calculations in pyCIAM and is thus outside of the scope of the current implementation.

## 4   Conclusion

Modeling the social and economic impact of future sea level rise can inform our understanding of costs in different climate change mitigation scenarios and support the analysis of adaptation policies. To construct global estimates, modelers face the dual challenge of developing a globally generalizable approach that is also capable of representing the detailed local information relevant to accurately estimating SLR impacts and adaptation. Prior modeling studies have developed frameworks for conducting such analyses; however, continued iteration of these data and models is necessary in order to improve the accuracy

and precision of projections and to keep pace with relevant advancements in data, modeling, and computing. Achieving this through community-wide collaboration requires a collection of open-source and transparent datasets as well as modeling tools.

This paper has summarized improvements to the quality and accessibility of both coastal impact data products and related modeling platforms. The Sea Level Impacts Input Dataset by Elevation, Region and Scenarios (**SLIIDERS**) represents a globally comprehensive and consistent collection of physical, ecological and socioeconomic variables for roughly 10,000 coastal

localities. SLIIDERS is a segment-wise data product for coastal impacts, similar to previous products like DIVA (Vafeidis et al.,

2008), but with significant improvements to the quality of represented variables. It is available as an open-source resource following FAIR guidelines (Wilkinson et al., 2016). Any researcher can download, inspect and alter SLIIDERS to utilize in their own coastal modeling studies.

The Python-Coastal Impacts and Adaptation Model (**pyCIAM**), a companion model that utilizes **SLIIDERS** as an input, was developed as an open-source update to the original Coastal Impacts and Adaptation Model Diaz (2016) and incorporates numerous improvements to model functionality and efficiency. pyCIAM is also made available as a modular, open-source tool meant to be modified by users seeking to add functionality or improve input sources, with users able to combine the model with their own input datasets, provided they are formatted similarly to SLIIDERS. An additional key advance of pyCIAM is that it is designed to simulate impacts from tens to hundreds of thousands of future SLR scenarios in parallel, facilitating scalable probabilisitic impact modeling research.

Results from pyCIAM v1.1, paired with SLIIDERS v1.1, show the model produces roughly similar estimates of the global net present cost of SLR to those of CIAM (Diaz, 2016) under the SSP5 socioeconomic scenario, with all other SSP-economic growth model configurations producing slightly smaller values (Fig. 4). Median annual, end of century costs under optimal adaptation in pyCIAM are also very similar to CIAM when averaging across all SSPs and growth models. When prohibiting proactive adaptation, costs are higher in pyCIAM for almost all scenarios as compared to CIAM. However, when comparing total yearly coastal damages, rather than just the climate-driven component, pyCIAM projects global NPV of all coastal damages between 2005-2100 to be roughly 3-4× those of CIAM (Fig. B1), likely due to greater population and capital stock estimates in these SSPs as compared to the trajectories used in Diaz (2016). The median annual, end of century total costs under optimal adaptation in pyCIAM are also higher than CIAM for all scenarios, with only the SSP4-IIASA scenario producing similar values.

Despite the improvements represented by the SLIIDERS data product and pyCIAM platform, there are aspects of them that should be improved in the future. We believe that a priority for future work should be to incorporate empirical evidence on coastal damages and adaptation behavior due to rising and extreme sea levels in order to better inform model assumptions. We hope that improvements to SLIIDERS can be made regularly as new, higher quality data sources for each of its constituent variables are made available. Additionally, the segmentation of coastlines in SLIIDERS v1.1 can be improved beyond a uniform (50km) spacing nested at the country level to better approximate coastal regions that behave as distinct decision making units, for example by capturing the extent of coastal urban centers. We intend to make many of these improvements moving forward, and will make updated versions available as such efforts are carried out. However, our hope is that the open-source nature of both SLIIDERS and pyCIAM will enable community-driven development to spur more rapid and substantial improvements to both tools.

## 5 Code and data availability

Version 1.1 of both the SLIIDERS dataset and pyCIAM model, is associated with the results presented in this manuscript. The SLIIDERS dataset, along with the code to create it, is available at https://doi.org/10.5281/zenodo.7693868. Source code for

SLIIDERS is also available at https://github.com/ClimateImpactLab/sliiders, where the 1.1 release corresponds to the version used in this manuscript and included in the Zenodo deposit. The model outputs used in this manuscript, along with the pyCIAM source code, are available at https://doi.org/10.5281/zenodo.7693869. Similarly, the pyCIAM source code is available at https://github.com/ClimateImpactLab/pyCIAM, with release 1.1 again corresponding to the model used for this manuscript. pyCIAM is also available on PyPI as the *python-CIAM* package. Scripts and notebooks associated with running pyCIAM and creating the results contained in this manuscript are also included in the pyCIAM GitHub repository and the Zenodo repository.

## Appendix A: Supplemental Information

### A1 Coastlines Creation and Length Calculation

To create each segment represented in SLIIDERS and used in pyCIAM, we assembled a set of polylines according to the following steps:

1. Downloaded highly-resolved 1:10m Natural Earth Coastlines (www.naturalearthdata.com/downloads/10m-physical-vectors).
2. Removed Caspian Sea borders from both coastline layers to avoid modeling along this inland sea.
3. Removed all line segments south of 60S (Antarctica) from both coastline layers to avoid inclusion of these coastlines in any final coastal segments, due to the lack of population and capital exposure any latitudes below 60S.
4. Converted coastlines layers to polygons in order to get land areas that correspond to the 1:10m scale coastline resolutions.
5. Intersected resulting polygon layer of land masses with exposure grid of population and capital assets, and removed land masses that contained no capital or population exposure. In these completely unpopulated areas, we cannot accurately represent value of lost land within the pyCIAM framework, nor is this value likely to be large.
6. Converted this 1:10m land area polygon layer back to polylines for use as our final vector layer of global coastlines.
7. Constructed a set of Voronoi polygons from the CoDEC-derived coastal segment centroids and intersected these with the coastlines layer constructed in Steps 1-8. This partitioned coastlines according to segment, allowing for the calculation of the total length (in kilometers) of coastline by coastal segment.

### A2 Aligning geographic and socioeconomic datasets to build SLIIDERS

Socioeconomic variables expressed in SLIIDERS and used in pyCIAM are defined at various geographic aggregation levels, from the fine "elevation bin by admin-1 region" scale to the coarse country scale. Input data sources also come in various formats, from gridded estimates of coastal elevation, population and capital distribution, and wetland area, to country-level SSP-based projections of income, population, and capital growth trajectories, to vector representations of country boundaries and coastlines. To create SLIIDERS, we must harmonize these various input sources. We start by assigning admin-1 and country labels to each grid cell in the gridded elevation and exposure input sources, using boundaries from GADM 4.1 (GAD). Notably, GADM uses the "country" label broadly, including many inhabited and uninhabited islands, regardless of sovereignty.

There are 211 countries in GADM 4.1 that are coastal and contain non-zero land under 20 m elevation. The boundaries of the admin-1 regions within these 199 countries are overlaid on gridded elevation and exposure datasets, including those defining spatial distributions of population (LandScan 2021) and physical capital (LitPop and GEG-15), to assign elevations and admin-1 labels to each grid cell. The gridded dryland and wetland area, population, and physical capital estimates are then binned by 10 cm elevation increments and grouped within admin-1 regions and coastal segments. Each admin-1 region is then assigned its corresponding country label, which is matched to the SSP-based country-level growth trajectories.

## A3    Estimating 2005-2020 capital stock values

Out of the 204 inhabited countries included in SLIIDERS, 143 have capital stock values from 2005 to 2019 in PWT 10.0 that we use as initial conditions for projecting capital stock consistent with the SSPs. We extend these one year using the Perpetual Inventory Method (PIM) and fill and/or impute the 61 remaining values using the following approach:

1. For 10 countries with ratios of non-financial wealth (NFW) to nominal GDP recorded in the 2022 Credit Suisse Global Wealth Databook (GWDB, Credit Suisse Research Institute (2022)) we use these ratios applied to previously gathered GDP estimates from PWT 10.0 and other sources and assume that the resulting NFW values are equivalent to physical capital stock, following the assumptions in Eberenz et al. (2020).

2. For 5 island departments of France, we use the NFW:GDP ratio from mainland France.

3. for 44 additional countries without individual estimates in the GWDB, we use regional averages, with regions defined by UNSTATS subregions (UNSD, 2021).

4. For North Korean estimates, we use capital:GDP ratios estimated in Pyo and Kim (2020) along with a Perpetual Inventory Method (PIM) parameterized by other parameters from Pyo and Kim (2020).

5. For Cuba, we take ratios of Cuban to U.S. captial stock from Berlemann and Wesselhöft (2017).

## A4    Projecting SSP-consistent (2020-2100) capital stock values

Using actual and imputed historical 2020 country-level capital stocks as initial conditions, we extract the capital portion of the OECD Env-Growth model (Dellink et al., 2017) and apply it to the SSP trajectories of GDP and population. The model requires global GDP elasticity of capital and 2020 country-level marginal products of capital (MPK), which are not described in Dellink et al. (2017). We use a global GDP elasticity of capital of 0.326 from Crespo Cuaresma (2017) and estimate 2010 MPKs using a modified Cobb-Douglas production function that contains only capital inputs. Coefficients of the function are derived by fitting to the compiled dataset of historical GDP and capital. Alternative approaches for obtaining these necessary inputs, including the use of a production function with labor and capital inputs and deriving the global elasticity directly from the production function, were also evaluated; however, these approaches yielded greater discrepancies in projected capital stocks when compared with the limited set of results presented in Dellink et al. (2017). To align most closely with citeEnvGrowth the aforementioned specification was chosen. The comparison of these alternative specifications is available in the SLIIDERS code repository accompanying this manuscript.

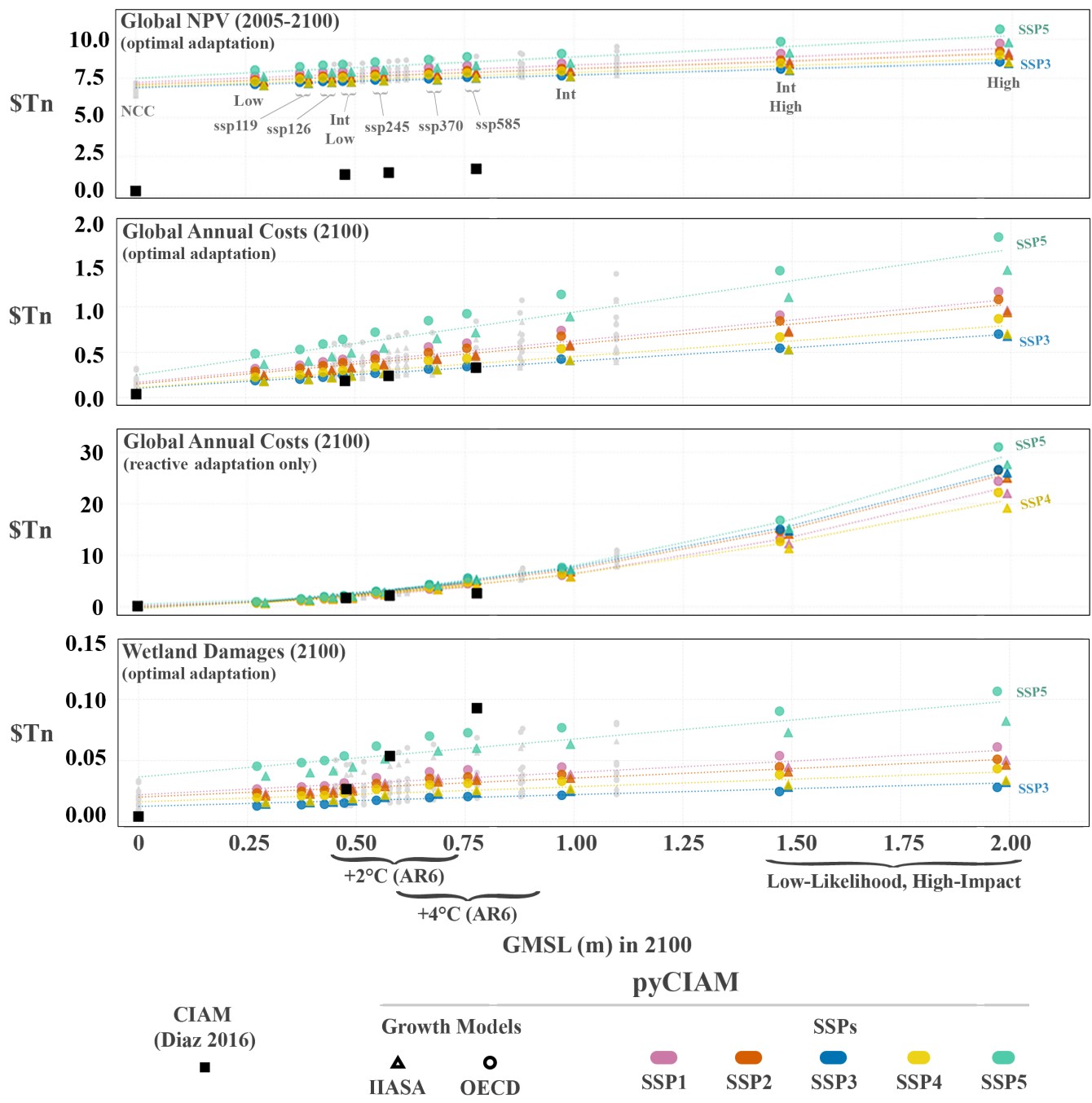

**Figure B1.** Comparison of global cost metrics for median model results under each SLR scenario. Values represent total coastal losses (inclusive of hazards not attributable to climate change), expressed in constant 2019 PPP USD. Each vertical group of points is a single SLR scenario, with each point in the group representing a unique combination of SSP-economic growth model. Differencing values associated with 0 GMSL rise from the other values yields Fig. 4. For visual clarity, only *medium confidence* AR6 and Sweet et al. (2022) scenarios are indicated with colored markers and jittered slightly along the x-axis based on runs using the OECD (-1cm) or IIASA (+1cm) economic growth model. The remaining SLR scenarios are shown in grey without jitter.

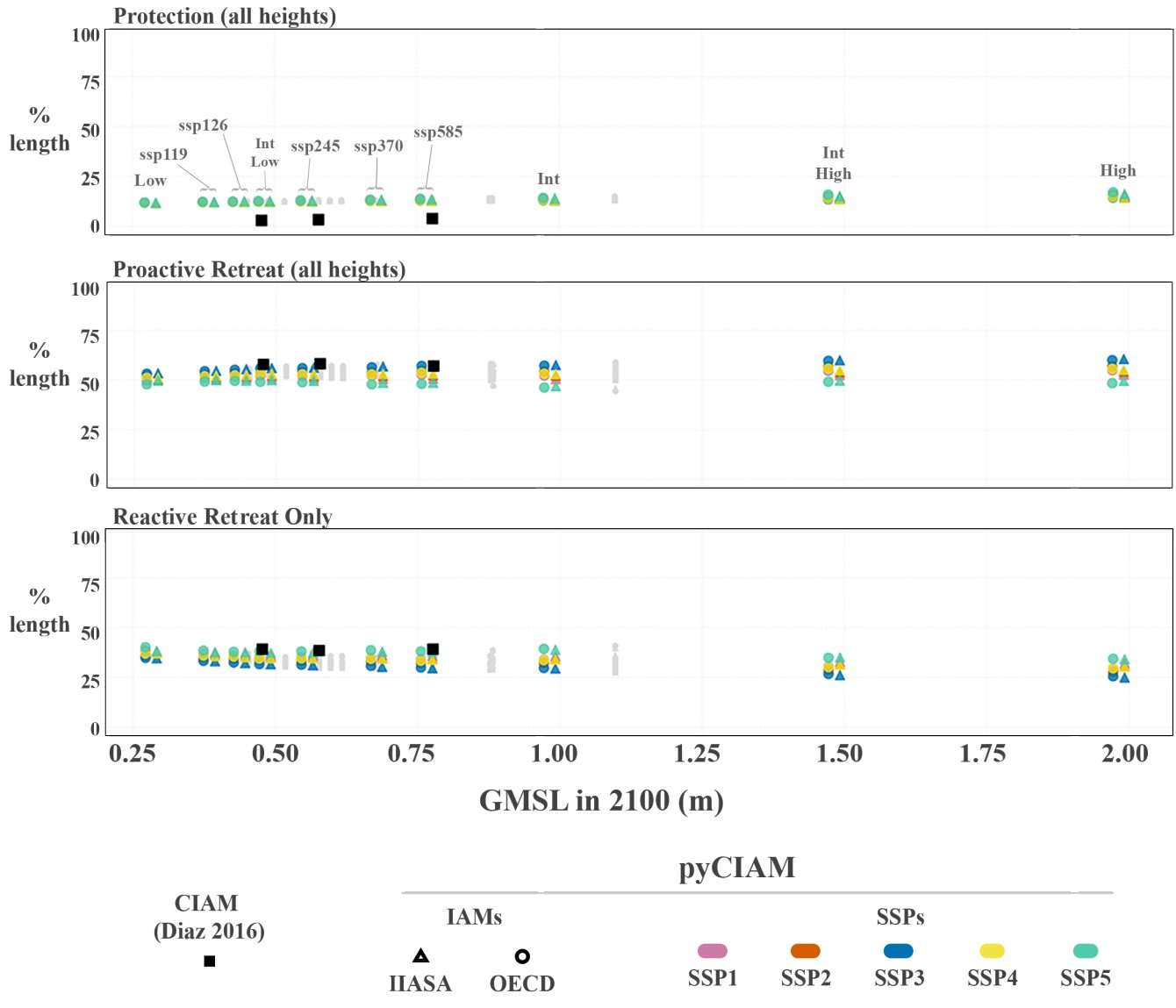

**Figure B2.** Comparison of optimal adaptation strategies adopted across all segments. Values represent percentages of global coastline associated with segments adopting each respective adaptation strategy. Proactive retreat and protection values are aggregates of all possible heights for each. Solid black squares represent the results from Diaz (2016). For visual clarity, only *medium confidence* AR6 and Sweet et al. (2022) scenarios are indicated with colored markers and jittered slightly along the x-axis based on runs using the OECD (-1cm) or IIASA (+1cm) economic growth model. The remaining SLR scenarios are shown in grey without jitter.

# Appendix C: Supplemental Tables

| ID | SLR Scenario | Model Used | GMSL in 2100 [m] (17th-percentile) | GMSL in 2100 [m] (83rd-percentile) |
|---|---|---|---|---|
| NCC | No Climate Change* | CIAM, pyCIAM | 0.00 | 0.00 |
| AR6-Med | IPCC AR6 *Medium Confidence* (2021) (SSP1-1.9, SSP1-2.6, SSP2-4.5, SSP3-7.0, SSP5-8.5) | pyCIAM | 0.28, 0.32, 0.44, 0.55, 0.63 | 0.55, 0.61, 0.76, 0.90, 1.02 |
| AR6-Low | IPCC AR6 *Low Confidence* (2021) (SSP1-2.6, SSP5-8.5) | pyCIAM | 0.32, 0.63 | 0.79, 1.61 |
| Sweet | US Interagency SLR Technical Report (2022) (Low, Int-Low, Int, Int-High, High) | pyCIAM | 0.28, 0.48, 0.98, 1.47, 1.95 | 0.29, 0.49, 0.99, 1.50, 2.02 |
| K14 | Kopp et al. (2014) (RCP 2.6, RCP 4.5, RCP 8.5) | CIAM, pyCIAM | 0.35, 0.43, 0.61 | 0.65, 0.76, 1.00 |
| SR | IPCC-SROCC (2019) (RCP 2.6, RCP 4.5, RCP 8.5) | pyCIAM | 0.39, 0.48, 0.71 | 0.60, 0.76, 1.11 |
| B19 | Bamber et al. (2019) (Low, High) | pyCIAM | 0.48, 0.79 | 0.96, 1.71 |
| D21 | DeConto et al. (2021) (RCP 2.6, RCP 4.5, RCP 8.5) | pyCIAM | 0.43, 0.52, 0.90 | 0.61, 0.74, 1.32 |

*Includes local background rates of relative sea level rise at each segment due to non-climatic background processes.

**Table C1.** GMSL rise between 2005 and 2100 for each 17th and 83rd percentile SLR scenario used in the pyCIAM and Diaz (2016) models.

| Input Dataset | Source & Description | DOI/URL |
|---|---|---|
| Coastal Segments | **CoDEC** (Muis et al., 2020): Defines segment centroids | 10.5281/zenodo.3660927 |
| | **Natural Earth 10m Physical Layers**: Defines global coastlines | https://www.naturalearthdata.com/downloads/10m-physical-vectors/ |
| Extreme sea levels (ESLs) | **CoDEC** (Muis et al., 2020) | 10.5281/zenodo.3660927 |
| Elevation | **CoastalDEM v2.1** (Kulp and Strauss, 2021): Primary elevation data source | https://assets.ctfassets.net/cxgxgstp8r5d/3f1LzJSnp7ZjFD4loDYnrA/71eaba2b8f8d642dd9a7e6581dce0c66/CoastalDEM_2.1_Scientific_Report_.pdf |
| | **SRTM15+ v2.5** (Tozer et al., 2019): Used to fill elevation data where CoastalDEM is undefined (e.g. polar latitudes) | 10.1029/2019EA000658 |
| | **MDT Global CNES-CLS18** (Mulet et al., 2021): Estimates of present-day mean sea level height relative to geoid | 10.5194/os-17-789-2021 |
| Wetland and Mangrove Extent | **GLOBCOVER v2.3** (European Space Agency and UCLouvain, 2010): Defines wetland extent | 10.1594/PANGAEA.787668 |
| | **Global Mangrove Watch 2016** (Bunting et al., 2018): Defines mangroves extent | 10.3390/rs1010669 |
| Local and global sea level rise projections | **LocalizeSL** (projections corresponding to Kopp et al., 2014; Bamber et al., 2019; Oppenheimer et al., 2019; DeConto et al., 2021): Local sea level rise projection outputs from the LocalizeSL model (used with AR5 emissions scenarios and other custom global temperature trajectories) | 10.5281/zenodo.6029807 |

| | |
|---|---|
| **Framework for Assessing Changes to Sea Level (FACTS)** (Fox-Kemper et al., 2021; Kopp et al., 2023; Garner et al., 2022; Sweet et al., 2022): Local sea level rise projection outputs from the FACTS model (used with AR6 and Sweet projections. | 10.5281/zenodo.6382554, 10.5281/zenodo.6382554 |

Table C3: Input data sources used to construct physical variables in SLIIDERS.

| Input Dataset | Source & Description | DOI/URL |
|---|---|---|
| Historical Population | **LandScan 2021** (Sims et al., 2022): Spatial distribution of global population in 2019 at a 30 arc-second resolution (~1km at equator) | 10.48690/1527702 |
| | **PWT 10.0** (Feenstra et al., 2015): Country-level time series of population | 10.34894/QT5BCC |
| | **UN World Population Prospects** (UN DESA, 2019): Used to fill population data for countries missing in PWT | https://population.un.org/wpp/Download |
| | **CIA World Factbook** (Agency, 2021): Used to fill population data for countries missing in PWT | https://www.cia.gov/the-world-factbook/ |
| | **World Bank World Development Indicators** (Bank, 2021): Used to fill population data for countries missing in PWT | 10.57966/6rwy-0b07 |
| | **Statistics and Research Åland** (Ala): Used to estimate population in Åland Islands | https://www.asub.ax/en |
| | **StatBank Norway** (Sta): Used to estimate population in Svalbard and Jan Mayen | https://www.ssb.no/en/statbank/table/07429 |
| Historical GDP | **PWT 10.0** (Feenstra et al., 2015): Country-level estimates of GDP per capita | 10.34894/QT5BCC |
| | **Fariss et al. (2022)**: Used to fill GDP data for countries missing in PWT | 10.1177/0022002721105443 |

| | | |
|---|---|---|
| | **World Bank World Development Indicators** (Bank, 2021): Used to fill GDP data for countries missing in PWT | 10.57966/6rwy-0b07 |
| | **IMF World Economic Outlook** (IMF, 2021): Used to fill GDP data for countries missing in PWT | https://www.imf.org/en/Publications/WEO/weo-database/2022/April |
| | **OECD regional statistics** (for Economic Cooperation and Development, 2020): Used to disaggregate French population into overseas departments | 10.1787/region-data-en |
| | **United Nations System of National Accounts** (UNSD, 2021): Used to fill GDP data for countries missing in PWT | https://unstats.un.org/unsd/snaama |
| Physical capital | **LitPop** (Eberenz et al., 2020): Gridded estimates of physical capital stock | 10.3929/ethz-b-000331316 |
| | **2015 Global Assessment Report (GEG-15)** (Bono and Chatenoux, 2014): Gridded of physical capital stock used to fill missing regions in LitPop | https://www.undrr.org/quick/11514 |
| | **PWT 10.0** (Feenstra et al., 2015): Country-level time series of capital stock estimates | 10.34894/QT5BCC |
| | **Credit Suisse Global Wealth Databook** (Credit Suisse Research Institute, 2022): Country-level time series of non-financial wealth used to fill capital stock estimates for countries missing in PWT | https://www.credit-suisse.com/about-us/en/reports-research/global-wealth-report.html |

| | | |
|---|---|---|
| | **Berlemann and Wesselhöft (2017); Pyo and Kim (2020)**: Estimates of capital stock:GDP ratios for select countries not contained in other sources | 10.1515/roe-2017-0004<br>10.1088/1748-9326/aaac87 |
| Mobile capital fraction | **PWT 10.0** (Feenstra et al., 2015): Capital is reported in PWT by category; structures are assumed to be immobile, with all other categories assumed as mobile | 10.34894/QT5BCC |
| Socioeconomic growth trajectories | **Shared Socioeconomic Pathways** (Riahi et al., 2017): Contains population projections from Kc and Lutz (2017) and GDP projections from Crespo Cuaresma (2017) and Dellink et al. (2017). We augment these with capital stock projections derived from the model defined in Dellink et al. (2017) | 110.1016/j.gloenvcha.2016.05.009<br>10.1016/j.gloenvcha.2014.06.004<br>10.1016/j.gloenvcha.2015.02.012<br>10.1016/j.gloenvcha.2015.06.004 |
| Construction cost indices | **World Bank ICP** (World Bank, 2020) | 10.57966/vm5h-a627 |
| | **Lincke and Hinkel (2021)** | 10.1029/2020EF001965 |

Table C4: Input data sources used to construct socioeconomic variables in SLIIDERS.

| SLR Scenario | Model | Socioecon. Scenario | GMSL [m] (2100) | NPV $Tn (bp) Optimal | NPV $Tn (bp) Reactive | Costs (2100) $Tn (bp) Optimal | Costs (2100) $Tn (bp) Reactive |
|---|---|---|---|---|---|---|---|
| RCP 2.6 | pyCIAM | SSP2/IIASA | 0.48 | 1.00 (2) | 2.94 (6) | 0.14 (3) | 1.67 (32) |
| *RCP 2.6* | *CIAM* | *IMF WEO* | *0.48* | *1.05* | *6.84* | *0.15 (8)* | *1.58 (92)* |
| RCP 4.5 | pyCIAM | SSP2/IIASA | 0.58 | 1.11 (2) | 3.80 (8) | 0.18 (3) | 2.53 (48) |
| *RCP 4.5* | *CIAM* | *IMF WEO* | *0.58* | *1.17* | *7.93* | *0.20 (12)* | *2.04 (118)* |
| RCP 8.5 | pyCIAM | SSP2/IIASA | 0.78 | 1.31 (3) | 5.83 (13) | 0.27 (5) | 4.68 (89) |
| *RCP 8.5* | *CIAM* | *IMF WEO* | *0.78* | *1.42* | *9.70* | *0.29 (17)* | *2.50 (145)* |

**Table C2.** Comparison of global estimated NPV (2005-2100) and annual costs of climate-driven SLR in 2100, expressed in constant 2019 PPP USD, between pyCIAM and Diaz (2016). Each metric is presented for both the optimal adaptation and reactive retreat modeling configurations. pyCIAM results are shown for the SSP2/IIASA socioeconomic growth scenario, while Diaz (2016) results are shown for the IMF World Economic Outlook (2011) projections used in that analysis. NPV for Diaz (2016) have been recalculated to be consistent with the 2005-2100 period used in pyCIAM. Numbers in parentheses show the fraction of global GDP associated with these costs in units of basis points (1/100ths of a percent). For columns 3 and 4, the NPV of GDP 2005-2100 is used for this calculation; for columns 5 and 6, GDP in 2100 is used. For Diaz (2016) scenarios, the 2100 global GDP used associated with the socioeconomic projections used in that analysis ($147.6 trillion 2010 USD) is reported in the paper. We use that value, adjusted to 2019 USD, to normalize the GDP impacts from Diaz (2016) scenarios. The NPV of GDP from 2005-2010 is not reported in Diaz (2016); thus we do not normalize Diaz (2016) NPV impacts.

*Author contributions.* Project conceptualized by IB, SH, REK. Data curation by DA, IB, JC, ND, AH. Methodology development, investigation, and formal analysis conducted by DA, IB, JC, ND. Map and figure visualizations were done by ND. Code base developed by DA, IB, JC, ND. Software developed by DA, IB, JC and ND. Model validation performed by IB and ND. Project administered by IB, ND, SH. Original draft written by IB, JC, DA, ND. Manuscript review and editing by DA, IB, JC, ND, SH, REK, MG. Funding acquisition by MG, SH, TH, REK. Supervision by MD, MG, SH, REK.

*Competing interests.* The authors declare that they have no conflict of interest

*Acknowledgements.* We thank Maya Norman for conducting her review and evaluation of relevant exposure datasets. Thank you also to Delavane Diaz, who contributed invaluable assistance in the access and interpretation of the CIAM model. We also thank members of the Climate Impact Lab who provided important feedback and guidance during frequent discussions about model objectives and developments. This project is an output of the Climate Impact Lab that gratefully acknowledges funding from the Energy Policy Institute of Chicago (EPIC), International Growth Centre, National Science Foundation (ICER-1663807), Sloan Foundation, Carnegie Corporation, and Tata Center for Development.

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
