# Peer review of "DSCIM-Coastal v1.1: An Open-Source Modeling Platform for Global Impacts of Sea Level Rise"

_EGUsphere, 2022_

## Author Response (AR1)

**DSCIM-Coastal v1.1: An Open-Source Modeling Platform for Global Impacts of Sea Level Rise**

The following discussion represents our consideration and response to the comments provided by the two reviewers of our manuscript. Overall, we were pleased to find the feedback positive and helpful. The majority of comments pertained to clarifications in wording, descriptions of methods or structure of the article, with a number of suggestions for improved visualizations in our figures. We have implemented the vast majority of these suggestions in our final revised submission. We are also in agreement with reviewer comments pertaining to potential input data improvements and have incorporated these updates where appropriate. Our responses to each comment are laid out in order of reviewer, with our responses provided in line. For each comment, our responses consist of a conversational reply to the reviewer, as well as a reference to the updated text or material in the revised manuscript itself that reflects the changes we made in response. A small number of comments provided by the two reviewers dealt with similar topics, for which our response text is similar. In response to the sentiments of the reviewers, we also implemented several minor updates to our model and manuscript to improve the rigor of analysis and interpretability of the results. In particular:

- We added to our analysis the median projections from IPCC's 6th Assessment Report (AR6), as requested by reviewer 2. We also added the five primary sea level rise scenarios detailed in the 2022 Inter-Agency Sea Level Rise Technical Report (Sweet et al. 2022), which complement the AR6 projections and allow us to estimate outcomes over a greater range of sea level rise. Together, these yield a total of 12 new scenarios in the updated model compared to the originally-submitted version.
- With the addition of new scenarios, we made slight changes to the abstract, end of the introduction and results sections. These changes are primarily in the annual costs results highlighted, to reflect three ranges of future sea level rise (SLR) and model outcomes under the AR6 and Sweet SLR projections. We have also contextualized the estimated costs by additionally presenting them as a fraction of global GDP.
- The nomenclature of the model input dataset (SLIIDERS) was simplified from "SLIIDERS-ECON" and "SLIIDERS-SLR" to simply "SLIIIDERS", which is equivalent to

the previous "SLIIDERS-ECON" dataset of physical and socioeconomic coastal characteristics. Originally, we had included SLIIDERS-SLR as a standalone data product as it represented the outputs of running the LocalizeSL sea level projection framework on a number of relevant scenarios. Upon adding the additional AR6 and Sweet SLR projections, which were more directly available without the need to run a 3rd party projection model and reflect more recent scientific advancements, we determined that SLIIDERS-SLR was no longer an important data product to release on its own. Further, we felt the dual data product context introduced unnecessary confusion. All text, dataset names and Figure 1 were altered accordingly, and we refer to this new version of "SLIIDERS-ECON 1.0" as "SLIIDERS 1.1."

- All sea level values are now expressed in relative terms to global mean sea level in 2005, rather than 2000 because the added AR6 scenarios begin in 2005 rather than 2000.

- For similar reasons, our results now reflect the pyCIAM model being run from 2005 to 2100, as opposed to 2000-2100.

- Given the questions and points raised by both reviewers about the current implementation of pre-existing protection (i.e. levees), and our realization of certain limitations of our original approach, we opted to alter the way in which initial, pre-existing protection is modeled. Instead of applying an "infinite protection" assumption to all areas protected by levees in the US and all areas in the Netherlands, we allow the pyCIAM model to choose the initial protection height for segments that reflect these same areas. This is identical to the approach taken in Diaz (2016) and ensures that the model's approach to estimating initial protection heights is consistent with its approach for estimating construction of future protection. We used a finer resolution version dataset of global coastlines to improve our estimation of segment coastline length. This, combined with an update of the subnational administrative unit geographic dataset used in the model (GADM), results in the total number of modeled segments increasing from 9,087 previously to 9,568.

- The main cost metric scatter plot in the results section (Fig. 4), which now contains 23 SLR scenarios, was formatted to better provide visual clarity between scenarios, SSPs and economic growth models, partly in response to comments from reviewers. We chose to highlight with colored markers just the *medium confidence* AR6 and Sweet SLR projections (n=10), which span the full range of end of century GMSL values across all 23 scenarios, with the remaining

scenarios' markers shown in gray. Additionally, because the addition of the Sweet scenarios allowed our results to span a broader range of sea level rise and therefore clearly reveal trends, trendlines were added to all scatter plots across these ten SLR scenarios by SSP (linear for all plots except global annual costs in 2100 under reactive adaptation alone, which required a quadratic fit), with their slopes (i.e. damage function values) indicated graphically as well. Mention of these values is also provided briefly in the results section text itself. Supplemental Figure B1 was updated similarly as well. Additionally, markers were added to these plots indicating the 2° C, 4° C and "High-End" GMSL ranges used in our results discussion (abstract, introduction, results) throughout.

- All maps (Figs. 5, 6, and 7) were updated to reflect results from the *medium confidence* AR6 scenario SSP2-4.5, using SSP2 and IIASA socioeconomic growth trajectories, in order to highlight a scenario from the most recent SLR projections.
- Table 3  was added, which presents global NPV and annual costs in 2100 under both optimal and reactive-only adaptation configurations for all *medium confidence* AR6 and Sweet SLR scenarios. Table C2 was also added, which for similar metrics shows a comparison between pyCIAM and Diaz (2016) results. Together, these replace the original Tables 4 and 5 in this section. We chose to do this to maintain consistency in showcasing results for a specific subset of contemporary SLR projections throughout the results text and figures, and to clearly delineate between (a) a comparison of our modeled results across the latest available SLR scenarios and (b) a comparison of our results to those of Diaz 2016 using a consistent set of SLR scenarios.
- We included a quantification of adaptation costs and benefits under what is now Section 3.2 "Adaptation Costs and Benefits" and included some of these findings in the abstract as well.
- Additional text was given minor edits for clarity while addressing reviewer comments. These changes can be seen in the revised submission that highlights all differences from the initial version.

Alongside these methodological and presentation updates, we updated numerous input data sources used in our paper because of releases of these data products that occurred in the interim period since we initially began the work. Due to the modular nature of the DSCIM-Coastal platform, these are easily ingested to generate updated results. In our

revised manuscript and analytical outputs (the DSCIM-Coastal platform), we will incorporate the following updates:

- CoastalDEM has been updated from v1.1 to v2.1, which is discussed in several of the responses below (see Kulp and Strauss, 2021 for details on this update)
- SRTM15+ (a low resolution global DEM used to augment our elevation information in areas where CoastalDEM does not exist) has been updated from v2.3 to v2.4
- Global Mangrove Watch 2016 has been updated from v2 to v3
- The UN World Population Prospects has been updated from the 2019 to the 2022 version
- The Asian Development Bank Key Indicators dataset has been updated from the 2021 version to the 2022 version
- The GADM dataset has been updated from v3.6 to v4.1
- The OECD Regional Statistics dataset has been updated from the 2021 version to the 2022 version
- LandScan has been updated from 2019 to 2021
- The Credit Suisse Global Wealth DataBook has been updated from 2021 to 2022
- The IMF World Economic Outlook was updated from October 2021 to April 2022

Implementing the above changes necessitated a re-run of the pyCIAM, such that all results, tables, figures, maps and related outputs in the manuscript and data products were updated to reflect these updates and alterations. This new model configuration was deemed "v1.1", as opposed to "v1.0", which is reflected in the manuscript title, text and related model/data products. Most of the impacts of these changes to the final results were small, however, such that the qualitative interpretation and discussion of results largely did not change. One value that shifted was the estimation of our "non-market costs of relocation" used by coastal segment agents to decide optimal adaptation strategies. This was previously approximated to be 5.0 times annual per capita income. Using the same approach but with updated data inputs, the new value of this ratio is estimated at 8.8 (Fig. 2). This entails a higher overall cost of relocation (i.e. coastal retreat), such that slightly more segments opt to build protection as opposed to undertaking proactive retreat. Previously the ratio of global populations opting to protect optimally was roughly 75% and retreat roughly 25%; these values are now closer to 80% and 20%, respectively.

**Responses to Reviewer Comments**

*Text in the following section is formatted as such:*

Normal Text: Reviewer's comments
Blue Text: Author's direct response to reviewer
*Blue Italics: Author's changes in manuscript in response to reviewer comment*

**R1 Comments and Responses**

This paper provides a framework, codes, data and results on future coastal impacts of sea-level rise under different adaptation scenarios. The paper represents a huge work, it is well presented and clearly written. It is excellent that the data, codes and framework are provided in a transparent manner. To me, the paper is definitively worth publishing, but some aspects require clarification from my perspective. I hope this review is useful.

***Moderate comments:***

Following the FAIR principles (Line 144) is excellent, but it is not demonstrated that the DSCIM platform is FAIR. One possibility could be to explain how the criteria of Force11 have been implemented.

We have included an additional footnote in the manuscript to better itemize the ways in which the DSCIM-Coastal platform abides by the FAIR principles according to the criteria detailed on Force11.org. This footnote can be found in the updated manuscript in **Section 1.3**:

*"Both components have been developed in accordance with FAIR Guiding Principles for scientific data management (Wilkinson et al., 2016) that are intended to improve the Findability, Accessibility, Interoperability, and Reuse of scientific data.\**
*\*[**FOOTNOTE**] These data and modeling components abide by the FAIR criterion as specified by The Future of Research Communications and e-Scholarship (FORCE11). Specifically, they are i) Findable via unique and persistent identifiers, with these identifiers specified in component metadata and indexed in a searchable resource (Zenodo, Github); ii) Accessible in that they are retrievable via these identifiers and are open, free and universally implementable; iii) Interpretable through the use of a formal, accessible, shared and broadly applicable language/vocabulary (manuscript and metadata in standard English and code in Python) and the inclusion of appropriate references to other data where necessary (e.g. input data sources); and iv) Reusable by specifying accurate and relevant attributes, applying an accessible data usage license and complying with coastal modeling community standards of language and data/code provision*

*(Force11.org)."*

It is not clear how flooding is modeled. Is this a bathtub approach? And in this case, given that the Bathtub approach generally highly overestimates the flooded area during events characterized by overflow, can this lead to overestimates of damage and adaptation costs? Similarly, how would the consideration of erosion and salinization of estuaries and coastal aquifers increase costs? I would not expect a detailed quantified value here but may be a note in the discussion.

Our localized sea level modeling approach accounts for large-scale heterogeneity in sea level rise and extreme sea levels experienced across segments, such that it is not a bathtub approach at the global scale; however, within each coastal segment, flooding is indeed modeled using a bathtub approach. We agree with the reviewer that implementing a more sophisticated hydrodynamic flood model for each segment could mitigate potential high-bias in flooding extents under the bathtub approach by accounting for land-surface roughness and flood deceleration dynamics. However, given the detailed input data and computational resources such an approach would require at the global scale, such an undertaking was deemed out of scope for this study.

It is also important to point out that our main findings concern the magnitude of climate-induced SLR-related damages. This refers to the difference between damages incurred under the various future emissions scenarios under climate change (CC) and those incurred under the no-climate change (no-CC) baseline scenario, both of which employ the same flood modeling approach. Therefore, general bias associated with our flood modeling (e.g. overestimation of flood extents) would be present in both the CC and no-CC trajectories and thus would be somewhat corrected when the CC vs. no-CC difference is calculated for each future CC scenario.

To add clarity to this component of our methodology, we have added the following sentences in **Sec 2.5.5 (Sea Level Rise)**:

*"…To estimate local sea level extremes, we linearly combine the fixed ESL distributions from CoDEC with an annually interpolated version of the decadal SLR projections from each of these 23 scenarios. This allows us to maintain a globally consistent representation of extremes at reasonably fine resolution. Limitations of this "local bathtub'' approach are described in Section 3.3."*

We have also included the following text to address the potential limitations in our flooding approach and with regard to the other processes highlighted in your review (e.g. erosion, salinization) in **Sec 3.3 (Model Limitations and Planned Improvements)**:

*"Our reflection of local mean and extreme sea levels is limited by the resolution of our local MSL projections (1 degree in FACTS, 2 degrees in LocalizeSL) and our ESL distributions from CoDEC (50km coastline spacing). Because of the desire to build a globally consistent model using these inputs, we employ a "local bathtub" model in which all points nearest to a given pair of ESL and MSL prediction points receive the same mean and extreme sea level projections. While this local model preserves the substantial large-scale spatial heterogeneity in SLR and ESL, sub-grid-scale variation is ignored. In particular, bathtub models are known to overestimate storm surge in inland areas largely due to the deceleration of flows caused by surface roughness (Bootsma 2022, Vousdoukas et al. 2016). A more sophisticated, dynamic representation of ESL based on local hydrodynamic simulations for each MSL/ESL combination is beyond the computational scope of this analysis but may yield improved future results and could be incorporated either "on-the-fly" within the pyCIAM model or in a pre-processing step that updates the ESL distributions in SLIIDERS.*

*Because pyCIAM linearly combines present-day ESL estimates and SLR predictions, our current approach also ignores changing ESL distributions due to (a) climate-driven changes to storm surge distributions from, for example, altered tropical cyclone frequency and intensity; and (b) the dynamic interaction between storm surge and MSL, moderated by local topography.*

*Despite these limitations in estimating sea levels, it is important to note that when isolating climate change-induced coastal costs, we difference the costs of a no-climate change baseline scenario that uses the same local bathtub flood model. This differencing also serves as a bias correction step, partially mitigating any over-estimates of flooding damages potentially introduced by the bathtub approach, though some high or low bias may still be present in the final results. Total (un-differenced) cost estimates (Fig. B1), however, will reflect any bias associated with the bathtub flood model. Accounting for these future changes is important for planning purposes, but represents a major computational challenge.*

*Additional geophysical dynamics associated with SLR inundation and related flooding, such as erosion, salinization of aquifers and estuaries, are also not currently addressed in our approach. Finer-scale wave setup and ESL behavior within complex coastlines at the sub-segment scale could also be useful to capture in future modeling. This would require estimates of ESLs at a much higher spatial resolution than is provided in the CoDEC dataset and is therefore currently infeasible given available input data."*

The resolution of the coastal segments is obviously a key issue: there is a trade-off between the computation time and the ability to remain realistic in terms of coastal extreme sea levels modeling. The study demonstrates well that reducing the number of points was possible because some regions representing a very small fraction of the total aggregated costs are simplified (e.g. French Polynesia). However I wonder to what extent the aggregated numbers given for 50km segments can be realistic. Typically, for example, the wave setup contribution to extreme sea levels is lower in harbors than in adjacent beaches (e.g., Lambert et al., 2020). What can be the impact of this simplification on the final results?

We agree that finer-scale modeling of coastal wave setup and ESL dynamics would be advantageous where possible. However, our ability to represent such processes is limited by the granularity of the outputs of  the global tide and surge model (CoDEC/GTSM) that generates our ESL values at each segment. These outputs are provided at 50 km spacing (10km in Europe, but we upscale to 50km for global consistency), which is often not sufficient for capturing finer-scale inlet/harbor related processes. The SLIIDERS framework is designed to be adaptable to future improvements in input data. Thus, increased resolution of global ESL estimates could easily be incorporated to better represent these processes in future iterations of the pyCIAM model. We have provided a sentence to this effect in **Sec 3.3 (Model Limitations and Potential Improvements)**:

*"Finer-scale wave setup and ESL behavior within complex coastlines at the sub-segment scale could also be useful to capture in future modeling. This would require estimates of ESLs at a much higher spatial resolution than is provided in the CoDEC dataset and is therefore currently infeasible given available input data."*

When discussing future socio-economic development, it is unclear why the work of Merkens et al. (2016) who downscaled SSPs in coastal areas is not acknowledged or discussed.

We appreciate the suggestion and have reviewed the downscaled coastal SSPs in the Merkens et al. (2016) paper (M2016). We compared the values in this dataset at the admin-1 level to those used in our current analysis, which scales present-day population distributions (from LandScan) with country-level growth projections. No consistent patterns emerged in the difference between our values and those underlying M2016. A

core objective of developing SLIIDERS was to update underlying data sources used in earlier iterations of similar datasets (e.g. DIVA). Our present-day population distribution is derived from 2021 population observations in the LandScan data product and thus contains more recent data than the Global Rural-Urban Mapping Project (GRUMP) dataset used in M2016. For this reason, along with the uncertainty and challenges associated with forecasting future subnational migration patterns, we chose to use the LandScan data layer, scaled over time by country-level population projections coming from the Institute for Applied Systems Analysis (IIASA), the official purveyor of the widely-used SSP database.

We believe that a benefit of creating our open-source platform is that it allows all interested future users to explore the differences in model results that would result from replacing our SSP population projections and distributions with those from M2016 or any other comparable data set. We now mention this possibility and cite M2016 in **Sec 2.2.5 (Extreme Sea Level Capital Damage)**:

*"However, should one wish to model within-country migration due to considerations such as SSP-consistent coastal urbanization and migration flows (e.g. Jones and O'Neill 2016, Merkens et al., 2016), such changes can be accommodated by updating the appropriate variables in the SLIIDERS input dataset."*

Figure 2 displays non-zero costs in states or regions which are not connected to the sea: e.g., Arizona in the US, Auvergne-Rhône-Alpes in France. In addition, it displays apparently zero costs in regions known to be highly exposed to sea-level rise (e.g. Occitanie in Mediterranean France, characterized by low lying urbanized sandy lidos). Can this be explained?

Non-zero costs in regions that are not hydraulically connected to the sea occurred in our initial analysis because we had not yet implemented a connectivity model to mask these areas out. Under the initial modeling approach, the "local bathtub" flooding approach used at each segment and described earlier in this response would flood these areas. To partially address this, we used the HydroSHEDS dataset to mask endorheic basins and additionally manually masked other known low-lying inland areas. However, such a process is not complete and missed some cases such as those you reference.

As a substantial improvement to our original implementation, and motivated by this comment, we have now incorporated a hydraulic connectivity model to mask areas that are not exposed to coastal extreme sea levels due to a lack of coastal connectivity. We have added this explanation to Section 2.5.3 - Elevation. We have also updated our topographic elevation dataset to utilize CoastalDEM v2.1 in place of v1.1 in order to integrate the significant improvements represented in that data update, as detailed in Kulp and Strauss (2021). This may have further mitigated the output of non-zero cost estimates in inland areas by improving the quality of elevation assignment and of the connectivity model.

Separately, the presence of zero costs in regions known to be highly exposed to SLR was due to a plotting error and was not reflected in the numerical model outputs. This has been corrected in the manuscript's figures, and we are again thankful for the careful review that uncovered this issue.

Section 2.5.3 now reads as follows:
*The use of accurate elevation data is crucial to appropriately representing sea level rise impacts (Kulp and Strauss 2019). We have implemented an updated elevation model used to define the population and physical capital exposed to SLR in pyCIAM in the following manner:*

1. *We utilize the CoastalDEM v2.1 dataset (Kulp and Straus 2021) to define elevations at 1 arc-second resolution (roughly 30m). The v2.1 release of CoastalDEM represents further improvements to the initially-released product (v1.1) (Kulp and Strauss 2018), though both datasets represent substantial accuracy improvements to prior DEMs, such as the widely used SRTM DEM. In addition to higher resolution elevation estimates compared to the 30-arc-second GLOBE DEM used in Diaz (2016), CoastalDEM significantly reduces bias found in SRTM, as presented in a comparative analysis based on CoastalDEM's initial release (v1.1) (Kulp and Strauss 2019). Compared to SRTM, CoastalDEM v1.1 suggests that roughly three times the amount of present day population resides below projected high tide levels under low emissions sea level rise scenarios by 2100 globally (Kulp and Strauss 2019). It should be noted that the high-resolution version of CoastalDEM v2.1 is the only input used in this study that is not publicly available. It is obtained via license with Climate Central, the developers of the DEM, though lower-resolution versions of the dataset are freely available for academic use. For the small number of regions that we model where CoastalDEM does not exist (e.g. above*

*and below 60N and 60S, respectively), we derive elevations from the SRTM15+ v2.5 dataset (Tozer et al. 2019).*

2. *We pair this DEM with 30 arc-second population estimates (Sims et al., 2022) and capital stock (LitPop, Eberenz et al. 2020) rasters, which allows for independent calculations of the distribution of land area, capital, and population with respect to elevation. We also rescale LitPop at the country-level to match more recently available data from Penn World Table 10.0 (Feenstra et al., 2015) and other sources (see Section 2.6.3). This approach differs from that of Diaz (2016), where population and capital stock densities were defined at the segment level and assumed to be homogeneously distributed within a segment.*

3. *We discretize the distributions of population and capital to 0.1m elevation slices, rather than 1.0m.*

4. *We mask all pixels that are not hydraulically connected to the ocean at 20 meters of SLR from analysis. This screens out most inland low-elevation areas not exposed to SLR. 20 meters is the highest elevation bin that we consider, reflecting the upper end of the ESLs that we consider combined with the upper end of local RSLR.*

We also address limitations of our current hydraulic connectivity model in Section 3.3:

*"...our hydraulic connectivity model masks only those regions that are not connected at 20 meters of SLR relative to 2005 levels. Some areas may not meet this criteria but still may be non-connected at lower sea levels. For example, a location that is at 1 meter above sea level but is behind a hill at 2 meters above sea level would be flooded by our model for sea levels of 1.5 meters. Future work could address this by assigning each pixel not only an elevation but a barrier height that would be treated similar to how manmade protection heights are treated in pyCIAM. This would increase the dimensionality of several calculations in pyCIAM and is thus outside of the scope of the current implementation."*

A diagram displaying the different components of the model and main principles would help the reader.

We agree and have included a new figure (FIG. 1) that replaces Table 1 at the end of **Sec 1.3**. We believe the figure better illustrates the hierarchy and structure of the various data and modeling components of the study:

**DSCIM-Coastal**
Open-source platform for computing global coastal impacts as part of the Climate Impact Lab's multi-sectoral Data-driven Spatial Climate Impact Model (DSCIM)

**Coastal Characteristics Dataset**
Sea Level Impacts Input Dataset by Elevation, Region, and Scenario for each coastal segment (**SLIIDERS**)

*Model Inputs*

**SLR Impact Modeling Platform**

**SLIIDERS**

**Physical Variables**
- Segment location and coastline length
- Land area by elevation (0.1m elevation bins)
- Extreme sea levels (10, 100, 1000, 10000-year)
- Wetland and mangrove area (0.1m elevation bins)

**Socioeconomic Variables**
Present (2019) and projected values by SSP-IAM:
- Population
- GDP (annual per capita income)
- Physical capital
- Construction costs

**pyCIAM**
Python-based Coastal Impacts and Adaptation Model

**Least-cost adaptation option for each segment**
One of the following:
- protect to a given extreme sea level (ESL) height
- retreat proactively to a given ESL height
- retreat reactively to local relative sea level rise (RSLR) alone

**Segment-wise costs (i.e. damages) outputs**
Under least-cost option and for reactive retreat only:
- Permanent inundation of land due to LSLR
- Wetland/mangrove loss due to LSLR
- Capital stock damage due to ESLs
- Population mortality due to ESLs
- Relocation (reactive and proactive retreat)
- Protective barrier construction

**Sea Level Rise Projections**
Local relative SLR projections from 23 distinct scenarios for different emissions pathways:
- IPCC Sixth Assessment Report ($n=8$)
  - 5 medium-confidence, 3 low-confidence
- U.S. Inter-Agency SLR Technical Report ($n=5$)
- IPCC Fifth Assessment Report ($n=3$)
- IPCC SROCC ($n=3$)
- Increased ice sheet instability scenarios ($n=5$)

**Minor comments:**

Line 114: please note that the IPCC glossary gives slightly different definitions for these terms (e.g., costs are a measure of some impacts)

We have updated this line as follows:
*"CIAM differentiated between six types of costs (i.e. "damages")..."*

Line 205: maybe add 'are characterized as follows..."
We have updated this line as follows:

*"Following Diaz (2016), pyCIAM separately tracks inundation costs, retreat costs, protection costs, cost of wetlands loss, and extreme sea level damage and mortality. These categories of costs are all used in cost minimization, and each is detailed below."*

Line 239-250: this is a reminder of the Method of Diaz, 2016, but another method is used here. Is this section necessary here?

This section is used to describe two key issues with the original (Diaz, 2016) implementation that motivate the different approach that we take. Thus, we feel that it is a relevant passage. However, to improve readability, we have now removed the detailed replication of equations presented in Diaz (2016) and replaced them with a reference to the equations in the original paper.

Line 256-257: I don't understand which population redistribution is meant here: is this due to SSPs or due to additional coastal migrations, e.g., in response to SLR?

This refers specifically to any non-SLR induced, within-segment redistribution across elevation slices. For example, if a segment's population was to migrate toward or away from low-lying regions of the segment, that would affect the segment's fractional losses incurred for a given extreme sea level. Such a case would be realized if we were to use the subnational SSP projections of Merkens 2016 or Jones and O'Neill 2016, as described earlier. We have updated this part of the text to add clarity as follows:

*"However, should one wish to model within-country migration due to considerations such as SSP-consistent coastal urbanization and migration flows (e.g. Jones and O'Neill 2016, Merkens et al., 2016), such changes can be accommodated by updating the appropriate variables in the SLIIDERS input dataset."*

Line 267-269: I understand that you have accounted for mortality as described in the Diaz paper, and not in its implementation. Is this correct? Eventually, the sentence could be made a bit clearer. Further, is this the population exposed to flooding (not ESL) which is considered to compute mortality?

Yes, that is correct. We corrected the storm (ESL-driven) mortality calculations to match the specification in the Diaz (2016) paper, and not the implementation discovered in the original CIAM code, which differed from the description in the paper. This mortality calculation pertains solely to mortality due to extreme sea levels (ESL) during storms and other high-water events. There is no mortality assumed with the permanent, gradual inundation due solely to local sea level rise. We have updated the text to add clarity:

*"In the implementation of Diaz (2016) (i.e. the original CIAM code), both the mortality assumption and the depth-damage function appear to have been used in conjunction, although the text of the Diaz (2016) paper states that the depth-damage function should only be used in the estimation of capital stock damage, not mortality. We therefore corrected this discrepancy in our implementation of ESL-driven mortality estimates in pyCIAM."*

Line 294: It is excellent to model this issue, but should this issue be framed only in terms of benefits and costs, or could there be also a general aversion to relocation motivated e.g. by optimism biases and attachment to specific location?

We agree. Optimism-bias and emotional attachment to a specific location are some of the motivations for relocation aversion we are referring to when we say "non-market costs of relocation". In line 284 of the original manuscript, when we state "non-pecuniary emotional consequences", we are implying precisely these sorts of aversions or attachments that likely impede relocation relative to what would be expected if one was to only consider simple market-driven cost-benefit equilibria. Assigning dollar values (i.e. 8.8x annual local GDP per capita) to these factors in our model is simply a way of estimating this non-market relocation resistance in units of cost (dollars) that are harmonized with the other cost types being modeled. This implies that the costs we estimate reflect the total welfare impacts, inclusive of the monetized value of these preferences. To better explain this, the first paragraph of Section 2.3 has been modified as follows:

*"In pyCIAM we introduce a calibration of non-market retreat costs based on observed patterns of settlement. Non-market retreat costs are those costs that are not directly visible to the market, but which nonetheless are incurred by individuals if they chose to relocate. For example, the non-pecuniary emotional cost associated with moving or the loss of social networks due to moving would both be non-market retreat costs. Accounting for these impacts would indicate that the total welfare impact of forced relocation is greater than simply the market costs associated with abandoning immobile capital. The existence of non-market relocation costs are thought to explain the observation that some patterns of coastal adaptation currently would not appear to be economically rational based on market costs alone (McNamara et al., 2015; Armstrong et al., 2016; Haer et al., 2017; Bakkensen et al., 2018; Hinkel et al., 2018; Suckall et al., 2018).  Using only market costs, least-cost optimization would indicate that many real-world populations should relocate or protect themselves, thus there must exist unobserved non-market costs that keep those populations in their current*

*locations. We leverage this observation to estimate the approximate magnitude of non-market relocation costs that would be necessary to explain current global settlement patterns."*

Line 326: I don't understand what is meant by data fidelity here. Can you rephrase or explain?
We updated this line for clarity:

*"We note that this approach is facilitated by the resolution of the input data represented in SLIIDERS."*

Line 340: Table 2 shows these times apply for an Apple MacBook Pro laptop with a 2.8 GHz Quad-Core Intel Core i7 processor and 16GB of RAM, but Table 2 is referred to much later. May be add it here to help the reader
Agreed. Table 2 (now Table 1) should appear right after this sentence at the end of Sec. 2.4. This was a LaTeX compiling bug and has been corrected in the updated manuscript.

Line 358-359: the note on resolution between parenthesis is based on the resolution of the initial dataset, but it can be clearer by just giving the grid cell or segment size.
These 1:50m and 1:10m resolutions are indeed based on the Natural Earth coastline input dataset used, but those are also the same spatial vector layers that we use to represent our model's coastlines when performing geospatial computations, such as calculating coastline lengths for each segment. Therefore, those inputs were not altered or converted to raster grids, meaning their native resolution specifications still apply in our modeling context. The '1:50m' and '1:10m' labels indicate the scale of the physical vector layers. In other words, in the 1:10m scale product, variations in coastline contours up to 10m should be reflected by the vector polylines in this product. One can think of these scale values as a form of smoothing of coastline shapes, where the value indicates the maximum length of coastline across which smoothing can occur. As such, 1:50m coastlines are more smoothed (less granular) than the 1:10m product. However, one additional update we have made for clarity and to capture more granular detail of global coastlines was to simply utilize the 1:10m Natural Earth coastline dataset for all coastal segments and have updated the text throughout the manuscript to reflect this change.

We have added a footnote in the manuscript at this location clarifying coastline scale values, in **Section 2.5.1** as follows:

*"The '1:10m' label indicates the scale of the physical vector layers, which can also be thought of as the maximum length of coastline across which simplification of complex coastlines into straight line segments can occur. 1:10m coastlines are the most granular product provided by Natural Earth."*

Section 2.5.3 can be misunderstood: as I understand it, the CoastalDEM is not independent of SRTM, but it improves SRTM by learning coastal features in the US where there is Lidar and improving in other regions. Can this be clarified?

We agree and have added some text to clarify this point as follows:

*"In addition to higher resolution elevation estimates compared to the 30-arc-second GLOBE DEM used in Diaz (2016), CoastalDEM significantly reduces bias found in SRTM, as presented in a comparative analysis based on CoastalDEM's initial release (v1.1) (Kulp and Strauss, 2019)."*

Section 2.7.4 : what protection standard is assumed beyond the US and the Netherlands? Is this 1:100?

No protection standard was assumed beyond the US and Netherlands due to the lack of comprehensive data on such infrastructure globally. In response to this discrepancy, we have removed all explicit prescription of present-day protections, instead allowing pyCIAM itself to estimate those protection standards like it does for future years. This also addresses the limitation that these explicitly prescribed protection areas could only be implemented with "infinite protection". In other words, in the current implementation there was no way to prescribe a specific protection standard to certain areas within a coastal segment; instead we removed them entirely from the analysis. Our updated approach aligns with that taken in Diaz (2016). Other approaches to estimating current protection standards globally do exist and were considered for this analysis, though none could be consistently implemented alongside the least cost optimization model that pyCIAM uses to assess future protection strategies. We discuss these other approaches and the limitation of not explicitly prescribing present-day protection in Section 3.3

*Better global data describing existing coastal protection infrastructure would improve the accuracy of pyCIAM. Spatially resolved data on constructed protection around the globe is sparse. To overcome this, some studies assume a certain level of protection as being present in all coastal regions, making stylized assumptions based on population densities and national*

*GDP (Sadoff et al., 2015). Other studies develop statistical models to empirically ground such relationships (Scussolini et al., 2016), and these have been incorporated in other global coastal adaptation models (Tiggeloven et al., 2020) and could be evaluated for use in future versions of pyCIAM. Further improvements to certain regions could also be made using protection data collected by Hallegatte et al. (2013) for 136 coastal cities.*

In section 3.3 – Model limitations: I suggest to include here some aspects raised above, such as the 50km coastal segmentation and the flood modelling approach (bathtub?). In addition, the analysis is purely economic, but there are other aspects that can be included, such as the feasibility of adaptation measures in 2070/2100, when resources and energy availability and costs may be completely different than today (See SPM of the WGII). Another limitation could be the consideration of salinization and erosion, as rightly noted in the introduction.

Thank you for these suggestions. We have incorporated such discussion as detailed in our responses to the related items in the 'Moderate Comments' above. We added a sentence about the feasibility of adaptation measures as well, as follows:

*"Additionally, the potential changing feasibility of both adaptation and accommodation measures in future decades, due to potential factors related or unrelated to climate change, like shifting supply chain and/or labor market dynamics, are not currently represented. These may prove to be relevant to society's capacity to effectively adapt in the future."*

In Figure 5 and in the similar figures in the annex, it is difficult to see the difference between the IAM and CIAM models because triangles, circles and squares are superimposed. Can this be improved? One possibility could be to display them on different column?

Yes, we agree with this comment and have implemented some slight jittering of the scatter plots based on the economic growth model for each SLR scenario's group of points in this and similar figures to improve interpretability of the different marker types.

***Data and codes***:

I have visited the codes without trying to extract all data run them on my computers (I am limited in terms of computer performance, data storage and time to run models!). It seems to me that the dataset lacks a bit of explanations (why two files, the smallest one seems to contain a lot of files of 0kB). Furthermore, it is not completely clear from the abstract why the framework and the codes are in two different deposits.

Thank you for the feedback. We have updated the README in the pyCIAM repository to provide additional clarity on running the model. The goal with providing code and data in multiple deposits was to (a) separate source code from data products, since different users may be interested in one or the other of these objects and (b) emphasize the separate, and modular nature of DSCIM-Coastal, which involves a data aggregation and harmonization effort to create a global dataset of coastal socioeconomic and physical characteristics (SLIIDERS) and a coastal sea level rise impacts model (pyCIAM) that ingests those datasets as input. However, we agree that the four separate zenodo deposits can create unnecessary confusion. Based on this feedback, we have combined source code and data products into a single deposit each for SLIIDERS and the pyCIAM model outputs. The source code in these deposits will represent a snapshot of the source code at the time of manuscript publication; however, we will additionally maintain the SLIIDERS source code and the pyCIAM source code as separate github repositories because they will receive separate, continued development after the manuscript is published.

I hope the review is useful

It was indeed, and we are thankful for the thorough review. Your comments have improved the clarity of our manuscript and robustness of our analysis.

**R2 Comments and Responses**

**Major comments:**

First, I was trying to get the model running on my PC, but did not manage to install the model. I believe that a user description would be helpful (e.g. how to install the model; how to run the model e.g. the command one should use to start the model, how can updated data be incorporated/run by the model and so on). As the main advancement of the model is the transparent and open-source provision of the data and code, the authors should try to make it as easy as possible for other scientists to get the model

running. Due to the difficulties in setting up the model – I am currently not able to evaluate the code and data provided within the manuscript. Further, I was wondering if the intention of the platform is also to regularly update data if new e.g. elevation data become available (Line 36/37 and Line 101/102). If so - how are the authors planning to update the data?

Thank you for the feedback. A user description is provided in the README files associated with constructing the **SLIIDERS** dataset (in the SLIIDERS repository) and running the **pyCIAM** model (in the pyCIAM repository), including step-by-step instructions. In the current version, the execution of the model occurs in a Jupyter notebook environment, rather than via a command line interface, with an example provided in the "run-pyCIAM-slrquantiles.ipynb" notebook. To make this clearer, we have (a) moved these step-by-step instructions to the top of the README file and (b) provided additional comments to make the execution of the notebook more intuitive. In addition, we will provide a Docker file containing an environment with all necessary packages to execute the model and replicate our results. The latter resource is under development

We have designed the DSCIM-Coastal platform to encourage regular updating of both the input data sources used to create the **SLIIDERS** dataset and the coastal climate impact modeling software (**pyCIAM**). We hope for this to evolve into a community effort and, as such, are not intending to take sole ownership of these tasks. However, as detailed at the top of this response, we have ingested updates to numerous input data sources that have changed to date since we initially began development of this model. This was performed by updating the notebooks associated with data acquisition, cleaning, and processing within the **SLIIDERS** code repository. To better describe this process to community members that may wish to contribute and/or update **SLIIDERS** or **pyCIAM**, we will add instructions on how to do so in the associated READMEs.

Second, I wonder if it is always necessary to point out how Diaz (2016) calculates certain parameters. It may be less confusing for readers to focus mainly on the new model version. I think it is very important to mention what is new in the paper, but I find it a bit confusing and overwhelming in parts.

We agree with this concern and have removed some of the more convoluted and potentially misleading or overwhelming portions of the Diaz (2016) methods summaries. Specifically we removed equations (1) and (2) from 'Section 2.2.5 - Extreme Sea Level Capital Damage' that reflected Diaz (2016)'s implementation to improve readability. We instead simply provide a reference to the equations in the original paper.

Third, it is not clear from the manuscript how coastal flooding is calculated. I guess the authors used a bathtub method, but this should be clarified in the manuscript.

Yes, it is a local bathtub model for each individual segment, with the flooding levels at those segments determined by our localized estimates of sea level rise and storm surge. This comment is similar to one provided by R1, for which we provided the following response and updated language in the text.

Our localized sea level rise modeling approach accounts for large-scale heterogeneity in sea level rise and extreme sea levels experienced across segments, such that it is not a bathtub approach at the global scale; however, within each coastal segment, flooding is indeed modeled using a bathtub approach. We agree with the reviewer that implementing a more sophisticated hydrodynamic flood model for each segment could mitigate potential high-bias in flooding extents under the bathtub approach by accounting for land-surface roughness and flood deceleration dynamics. However, given the detailed input data and computational resources such an approach would require at the global scale, such an undertaking was deemed out of scope for this study.

It is also important to point out that our main findings concern the magnitude of climate-induced SLR-related damages. This refers to the difference between damages incurred under the various future emissions scenarios under climate change (CC) and those incurred under the no-climate change (no-CC) baseline scenario, both of which employ the same flood modeling approach. Therefore, general bias associated with our flood modeling (e.g. overestimation of flood extents) would be present in both the CC and no-CC trajectories and thus would be somewhat corrected when the CC vs. no-CC difference is calculated for each future CC scenario.

To add clarity to this component of our methodology, we have added the following sentences in **Sec 2.5.5 (Sea Level Rise)**:

*"…To estimate local sea level extremes, we linearly combine the fixed ESL distributions from CoDEC with an annually interpolated version of the decadal SLR projections from each of these 23 scenarios. This allows us to maintain a globally consistent representation of extremes at reasonably fine resolution. Limitations of this "local bathtub'' approach are described in Section 3.3….each region of the world experiences the median projected RSLR for that scenario. In physical terms, this means that the inundation and flooding experienced by a given segment is represented as a simple bathtub model, conditional on that segment's RSLR and ESL heights. However, it is important to note that, using the methods detailed in Kopp et al. 2014, 2017 and Muis et al. 2020, the heterogeneity of RSLR and ESL values across different segment locations is preserved in our approach."*

We have also included the following text to address the potential limitations in our flooding approach and with regard to the other processes highlighted in your review (e.g. erosion, salinization) in **Sec 3.3 (Model Limitations and Planned Improvements)**:

*"Our reflection of local mean and extreme sea levels is limited by the resolution of our local MSL projections (1 degree in FACTS, 2 degrees in LocalizeSL) and our ESL distributions from CoDEC (50 km coastline spacing). Because of the desire to build a globally consistent model using these inputs, we employ a "local bathtub'' model in which all points nearest to a given pair of ESL and MSL prediction points receive the same mean and extreme sea level projections. While this local model preserves the substantial large-scale spatial heterogeneity in SLR and ESL, sub-grid-scale variation is ignored. In particular, bathtub models are known to overestimate storm surge in inland areas largely due to the deceleration of flows caused by surface roughness (Bootsma 2022, Vousdoukas et al. 2016). A more sophisticated, dynamic representation of ESL based on local hydrodynamic simulations for each MSL/ESL combination is beyond the computational scope of this analysis but may yield improved future results and could be incorporated either "on-the-fly'' within the pyCIAM model or in a pre-processing step that updates the ESL distributions in SLIIDERS.*

*Such an approach would also address a second limitation in our local sea level estimation. By linearly combining present-day ESL estimates and SLR predictions, our current approach ignore changing ESL distributions due to (a) climate-driven changes to storm surge distributions from, for example, altered tropical cyclone patterns; and (b) the dynamic interaction between storm surge and MSL, moderated by local topography.*

*Despite these limitations in estimating sea levels, it is important to note that when isolating climate change-induced coastal costs, we difference the costs of a no-climate change baseline scenario that uses the same local bathtub flood model. This differencing also serves as a bias correction step, partially mitigating any over-estimates of flooding damages potentially introduced by the*

*bathtub approach, though some high or low bias may still be present in the final results. Total (un-differenced) cost estimates (Fig. B1), however, will reflect any bias associated with the bathtub flood model.*

*Finally, additional geophysical dynamics associated with SLR inundation and related flooding, such as erosion, salinization of aquifers and estuaries, are also not currently addressed in our approach."*

**Minor comments:**

It would be helpful to have a graph that gives an overview of the different modules within the model and input data.

We agree and have included a new figure (FIG01) that replaces Table 1 at the end of **Sec 1.3**. We believe the figure better illustrates the hierarchy and structure of the various data and modeling components of the study.

**DSCIM-Coastal**
Open-source platform for computing global coastal impacts as part of the Climate Impact Lab's multi-sectoral Data-driven Spatial Climate Impact Model (DSCIM)

*Coastal Characteristics Dataset*
Sea Level Impacts Input Dataset by Elevation, Region, and Scenario for each coastal segment (**SLIIDERS**)

*Model Inputs*

*SLR Impact Modeling Platform*

**SLIIDERS**

**Physical Variables**
- Segment location and coastline length
- Land area by elevation (0.1m elevation bins)
- Extreme sea levels (10, 100, 1000, 10000-year)
- Wetland and mangrove area (0.1m elevation bins)

**Socioeconomic Variables**
Present (2019) and projected values by SSP-IAM:
- Population
- GDP (annual per capita income)
- Physical capital
- Construction costs

**pyCIAM**
Python-based Coastal Impacts and Adaptation Model

**Least-cost adaptation option for each segment**
One of the following:
- protect to a given extreme sea level (ESL) height
- retreat proactively to a given ESL height
- retreat reactively to local relative sea level rise (RSLR) alone

**Segment-wise costs (i.e. damages) outputs**
Under least-cost option and for reactive retreat only:
- Permanent inundation of land due to LSLR
- Wetland/mangrove loss due to LSLR
- Capital stock damage due to ESLs
- Population mortality due to ESLs
- Relocation (reactive and proactive retreat)
- Protective barrier construction

**Sea Level Rise Projections**
Local relative SLR projections from 23 distinct scenarios for different emissions pathways:
- IPCC Sixth Assessment Report ($n=8$)
  - 5 medium-confidence, 3 low-confidence
- U.S. Inter-Agency SLR Technical Report ($n=5$)
- IPCC Fifth Assessment Report ($n=3$)
- IPCC SROCC ($n=3$)
- Increased ice sheet instability scenarios ($n=5$)

Line 72: What is meant by high-resolution here?

We altered this sentence to now read as:
*"Several past studies employed global coastal impact models to estimate future damages from SLR and ESLs under various trajectories of global GHG emissions, socioeconomic scenarios, and adaptation pathways for thousands of sub-national coastline segments (Hinkel et al., 2014; Diaz, 2016; Lincke and Hinkel, 2018, 2021)."*

Line 86: The DIVA modeling framework is published under the INSERT LICENSE. The code and data are available in the (protected) DIVA repository: https://gitlab.com/daniel.lincke.globalclimateforum.org/diva

We are aware of this landing page but have found that as of 3 Feb 2023 there is not actually a license listed, instead just the placeholder 'INSERT LICENSE' text. We checked the provided gitlab link's functionality both during the writing of the original manuscript (12 Apr 2022) as well as during this revision process (3 Feb 2023) and have not found it to be functional. We communicated this issue with the repository's creator but did not receive a response. There is a similar (unprotected) repository named "diva_published" located at https://gitlab.com/daniel.lincke.globalclimateforum.org/diva_published, which contains a variety of csv files and shapefiles that look to be input data to the creation of the original DIVA dataset. However, it does not contain any of the code used to create the DIVA dataset.

Line 138: Subheading title: 'The Data-driven Spatial Climate Impact Model Coastal Impacts Architecture' Is there a little mistake? It reads a bit strange to me.

Agreed. We have renamed this subheading '*This Study: The Data-driven Spatial Climate Impact Model - Coastal Impacts*'

Line 153: A table of the datasets that are included in Sliders and the references would be helpful for readers

Two such tables are already included in the appendix as tables C3 (physical variables) and C4 (socioeconomic variables). We opted against including these in the main text due to their size.

Line 155: Would it make sense to also include the new IPCC SLR scenarios (AR6)?

We agree that including these latest SLR scenarios would be a worthwhile inclusion. The bulk of our model creation and manuscript writing was done prior to the release of the finalized AR6 scenarios, but now that they are available we have integrated them into this platform. We have included seven total scenarios from AR6 (2 *low-confidence* and 5 *medium confidence*) (Fox-Kemper et al. 2021). We have also included the five primary scenarios detailed in the 2022 Inter-Agency Sea Level Rise Technical Report (Sweet et al. 2022) to complement the AR6 projections and add additional contemporary scenarios. The full suite of SLR scenarios is now 23 unique projections, combined with five SSPs and two economic growth models, for a total of 230 total future SLR trajectories. The pyCIAM model was re-run with all of these scenarios, with results, figures and tables throughout the text updated accordingly. We appreciate the suggestion; this ability to continuously update relevant input datasets as they are improved over time is one of the primary motivating factors for designing the SLIIDERS/pyCIAM platform in a modular, open-source fashion.

Line 188: How are people spatially relocated to an unaffected area within the model?

They are relocated to what is assumed to be a safe inland area not in danger of future sea level rise or coastal storm surge impacts - effectively removed from the model moving forward. We have added some clarity to this point in Section 2.1 as follows:

*"- Reactive Retreat: When a portion of land falls below MSL, all people and mobile capital are relocated to an unaffected, inland region away from the coast that is not in danger of future impacts from SLR or ESLs, and immobile capital is abandoned.*

*...*

*- Proactive Retreat: All people and mobile capital below a certain retreat height are assumed to be relocated to a safe, inland region, and immobile capital below that height is abandoned. The options for that retreat height level are discretized to the same values available for protection, with the addition of a 'low retreat' option representing the maximum MSL projected during a 'planning period' (10 years)."*

Line 255: What are the baseline costs?

In this context, baseline costs refers to the initial cost of constructing the protection. We have updated this sentence to read as follows:

*"As in Diaz (2016), maintenance costs are assumed to be 2% of the initial construction cost..."*

Line 265: 'VSL-valued' – What does VSL stand for?

We have altered the original text to provide clarity for this term, where VSL indicated 'value of a statistical life', as well as a citation for the paper from which our VSL framework was derived, the new sentence reads as follows:

"*This category reflects the expectation of annual costs of mortality occurring due to ESL events, where death equivalents are valued using a Value of a Statistical Life (VSL) framework, as employed in Diaz (2016), which assumes 1% mortality for all populations exposed to a given ESL, based on Jonkman and Vrijling (2008).*"

Line 267: Small typo: double 'of'

Corrected.

Line 275: Some more detail on how the height of adaptation measures is influenced/calculated would be interesting here.

We slightly altered this sentence to add clarity as follows:

*"The maximum heights of projected RSLR at each segment during a given planning period in turn influence the heights at which protect or retreat adaptation options are employed. For segments that adapt via reactive retreat, the height of retreat exactly matches this projected RSLR, while segments employing 10, 100, 1000, 10000-year retreat or protect actions consider the heights of these ESLs atop this projected RSLR baseline for that planning period."*

Line 327: DIVA assumes a homogenous population per elevation increment and segment

We re-worded this sentence to more clearly articulate the shortcoming of DIVA being its assumption of uniform population and capital densities across entire segment areas, while our approach leverages improved input datasets to disaggregate these densities by segment-elevation bin:

*"We note that this approach is facilitated by the resolution of the input data represented in SLIIDERS. The DIVA inputs used in Diaz (2016) assume that population and capital density are homogeneously distributed throughout each segment, and are non-varying by elevation."*

Section '2.5 Physical Model Inputs in SLIDERS': It might be useful to have this section first to understand the data model/structure better.

While we appreciate this suggestion to improve readability of this section, we feel that the current structure is necessary to maintain in order to preserve the logical flow of our work. Given that our work is heavily based on a pre-existing modeling platform's (CIAM) structure, we feel it is important to provide the context of that model's structure early on in this section, as the characteristics and limitations of CIAM were largely what compelled the type of updates and improvements we chose to make to the physical and socioeconomic input data. Therefore, we feel that detailing those updates subsequently to the model structure and related sections is a sensible layout.

Line 350ff: Why did you use 50km (and not e.g. 20km)? What is the reasoning behind it? If one thinks about adaptation units, a homogeneous segmentation might not be the best solution as parameters as population density, coastal geomorphology changes quite rapidly. For a global application, one must make simplifications, but I wonder how the authors come up with a 50km segmentation.

We used 50km segmentation because that was the coastal spacing between the majority of the global CoDEC Global Tide and Surge Model (GTSM) dataset (Muis et al. 2020), from which we drew our surge height values. Given the lack of modeled surge height at higher spacing and the computational constraints associated with growing the total number of coastal segments to be optimized for adaptation action, we deemed the 50km value to be suitable for this study. Utilizing smaller coastal segments would

not alter the aggregate losses projected in our model unless we were able to obtain similarly fine resolution input data describing either extreme sea levels or local sea level rise.

Section Elevation: Might be good to include several DEMs (in the long run) as the model results are very sensitive to variations in input data (e.g. see Hinkel et al. 2021)

We appreciate this comment and may consider this in future updates to the model moving forward. However, provisioning multiple versions of the SLIIDERS dataset and associated pyCIAM results using different DEM products was deemed out of the scope of the present study. This is partly due to the added complexity associated with this approach but mostly due to the fact that there is an increasing consensus that the CoastalDEM products represent the best data product for modeling coastal topography that is presently available. Along these lines, in this revision we have improved our DEM approach by replacing CoastalDEM v1.1 with the newly-developed CoastalDEM v2.1 (Kulp and Strauss 2021), which significantly improves upon version 1.1. Other members of the coastal impacts research community may utilize this framework to explore the impact of using alternative DEMs (or other input data sources, such as population or physical capital). Given the large number of datasets used as inputs in the creation of SLIIDERS, assessing the sensitivity to alternative sources is outside the scope of this model description paper; however, it may be the subject of future analyses.

Section 2.74: the FLOPROS database by Scussolini et al., 2016 is a first collection of current protection levels. Might be good to mention here.

Thank you for pointing out this study. We were aware of this work and view it as a potentially compelling avenue for further research. Interested users could potentially integrate the modeled protection levels from FLOPROS in an altered configuration of SLIIDERS and pyCIAM. However, given the fact that these protection levels are incomplete, largely based on policy rather than direct observation, and assume constant protection standards at state/province or sometimes country level, we opted to exclude them from the initial implementation of our modeling platform. A modeled layer from FLOPROS, in which design standards for areas without direct observation or policy data are imputed, exists for riverine flooding but not coastal flooding.

Additionally, CIAM (and pyCIAM) are not currently set up to incorporate prescribed initial protection standards. Rather, the model estimates its own initial design standards, like it does for subsequent time steps, based on a least cost analysis. This provides an alternative to an estimated design standard from a dataset like FLOPROS, but one that is consistent with the parameters used throughout the pyCIAM model. Nevertheless, we acknowledge FLOPROS utility and briefly highlight the potential for integration in future versions of the pyCIAM model in Section 3.3::

*"Other studies develop statistical models to empirically ground such relationships (Scussolini et al., 2016), and these have been incorporated in other global coastal adaptation models (Tiggeloven et al., 2020) and could be evaluated for use in future versions of pyCIAM."*

Line 545-545: This part is not clear to me. Could you please support the argument with references and explain your correction/improvement more precisely?

Because of the inconsistencies associated with prescribing present-day protections, as described in the previous comment, we have updated the model to no longer prescribe full protection to areas designated as within a levee protection area by the USACE National Levee Database (NLDB). Instead, we allow pyCIAM to assign an initial protection height associated with the least-cost design standard. This is consistent with the approach used in Diaz (2016). Because of this, we have removed this part of the text. As described in the previous response, we also include a brief discussion in Section 3.3 of the limitations of not prescribing present-day protection standards, along with highlighting a few candidates for providing this information in future work.

Figure 6, lowest panel: What is shown by the gray color (local sea level)?

This indicates that the retreat height undertaken by segments that adopt only "reactive retreat" as their adaptation strategy is equal simply to the local sea level at each segment. In other words, each of those segments retreats exactly in line with local sea level rise at that segment to avoid permanent inundation, but no higher.

---

## Author Response (AR2)

**DSCIM-Coastal v1.1: An Open-Source Modeling Platform for Global Impacts of Sea Level Rise**

The discussion below represents our response to the second round of reviewer comments, as facilitated by the topical editor following our revised and resubmitted manuscript after the initial round of feedback from Reviewers 1 and 2. Given the extent of our revisions, updates to input data and modifications to the original manuscript text, the editor deemed it necessary to solicit a final iteration of review by Reviewer 1. We agree and appreciate the careful review this publication has received. We hope the responses we provide here offer the clarity and detail sought by these most recent inquiries. We have not revised our manuscript further as we do not believe any of the reviewer's comments, nor our responses, suggest that changes are needed.

1. In the result section, the high costs in regions whose coasts are not densely populated such as Yakutia (Sakha republic) in North-east Russia are a bit surprising and may be worth double checking.

   This is a good observation and brings up an important characteristic of losses projected by pyCIAM. Not all cost types scale directly with population or capital density. The end-of-century annual costs incurred in the large region of Yakutia for the SSP2-4.5 (medium confidence), IIASA AR6 scenario shown in Fig. 5, totaling roughly $1.1 billion, are primarily driven by wetland loss and associated ecosystem services loss (valued at ~$900 million). This is due to the large extent of land area and abundance of low-lying, coastal wetland-classified lands within it. The next most significant cost type in this region is the value of (dry) land permanently lost to inundation ($80 million). Given that neither of these costs are dictated by the presence of capital assets or human population, it is logical that they are driving the cost signal seen here and in other sparsely-populated regions. The absence of capital and population in regions like Yakutia is reflected in the following figure, Fig. 6, which shows the benefits of optimal adaptation. Here, the values are near-zero for Yakutia and most high-latitude regions given the low amount of protection construction or proactive retreat occurring in these areas due to their low populations.

   Globally speaking, the sum of human-associated annual costs from storm damage, mortality, cost of building protection and cost of relocating are significantly larger than the combined wetland loss or land inundation costs. Here is the breakdown of global cost shares by cost type for annual costs in 2100 for that same scenario, assuming optimal adaptation:

   Total Costs (2100): $362 billion
   Wetland: 7.95 %

Inundation: 16.48 %
Relocation: 17.93 %
Protection: 32.67 %
Storm damage: 4.62 %
Storm mortality: 20.35 %

2. It would be good to know the levels of protection selected by the model in places such as the Netherlands, in order to compare with existing standards (which can indeed not be strictly followed on the ground).

In all scenarios, including in the no-climate-change scenarios, all 31 segments in the Netherlands adapt by building protection to at least the 1000-year surge height, with 21 of 31 segments adapting to the 10000-year height. This appears to reasonably approximate current protection in the country given the fact that the design heights of current protective infrastructure for all Dutch coastlines reported in the FLOPROS dataset (Scussolini et al., 2016) are listed as being between 4000-year and 10000-year surge heights.

3. It would be good to investigate the implementation of a GIS approach to limit the maximum extent of flooding in extremely low lying areas and avoid the overestimation of losses due to the bathtub approach.

We agree that improving upon the current implementation of our "locally bathtub" flood modeling approach is a good candidate for future improvements. We do currently employ an extensive amount of geospatial (i.e. GIS) processing to the CoastalDEM tiles in all regions in order to ensure that we are only considering areas below 20m that are hydraulically-connected to the ocean and therefore likely vulnerable to SLR. However, given the computational intensity and lack of relevant datasets (e.g. surface roughness, soil porosity etc.) that would be necessary for a more sophisticated hydrodynamic flood modeling approach at a similar (10m) resolution to our DEM, we opted to stick with the bathtub method for this v1 model release.

4. I noticed the 50km resolution and so does the other reviewer - I think that investigating ways to improve the resolution would be very relevant.

We agree and hope to improve upon this resolution in future iterations of this modeling platform. However, we are currently limited by the spatial granularity of the necessary input datasets available to us for this estimation.  Given the 50-km resolution of the CoDEC extreme sea level dataset, the gains from using smaller coastal segments would be limited because each of smaller segments within a 50km segment would be assigned identical flood heights. Furthermore, we are not aware of substantial evidence indicating the granularity at which protection or retreat decisions are made. Assigning too fine

resolution for the segments would enable overly flexible decision making beyond what is practical. Investigating methods to improve upon this resolution and the coastal segmentation algorithm in general is a consideration for us moving forward.

References:

Scussolini, P., Aerts, J.C.J.H., Jongman, B., Bouwer, L.M., Winsemius, H.C., De Moel, H. and Ward, P.J., 2015. FLOPROS: an evolving global database of flood protection standards. Nat. Hazards Earth Syst. Sci. Discuss, 3, pp.7275-309.

---

## Author Response (AR3)

**DSCIM-Coastal v1.1: An Open-Source Modeling Platform for Global Impacts of Sea Level Rise**

Dear Dr. Phipps,

Thank you for the opportunity to revise our paper further to incorporate our responses to Dr. Le Cozannet's second round of comments. In particular, you asked for two responses to be incorporated.

First, you suggested that we include text highlighting to readers that inundation costs in the model are a function of parameters beyond population, such that it is possible to experience non-zero costs in unpopulated regions. We agree that the lack of clarification regarding the CIAM methods for estimating the value of lost, potentially unpopulated, land to permanent inundation from SLR warranted attention. To address this, we have added the following paragraph in Section 2.2.1 (Inundation Costs):

> *Value of land lost permanently to inundation is estimated in accordance with the Diaz (2016) methodology, which approximates land values based on country-level assumptions of non-coastal land value from the integrated assessment model FUND (Tol, 1996). We assume that these national land values appreciate over time as a function of projected per capita income and population density growth in future years for each country. Equation 7 of Diaz (2016) Supplemental Information details the total cost of inundation as a function of land values and immobile capital loss. Because pyCIAM, like CIAM, estimates land to have value even in unpopulated regions, non-zero inundation costs are still incurred in unpopulated segments due to lost land (or lost wetland area, see Sec. 2.2.4), despite the absence of any immobile capital losses. As expected, the magnitudes of these inundation losses tend to be much lower than in highly populated segments exposed to SLR.*

Second, you suggested that we address Dr. LeCozannet's comments that "investigating ways to improve the resolution would be relevant" that "it would be good to investigate the implementation of a GIS approach to limit the maximum extent of flooding in extremely low lying areas and avoid the overestimation of losses due to the bathtub approach."

Here, our original response indicates that we do indeed implement a GIS approach, overlaying multiple raster and vector datasets at various resolutions in combination with our "local bathtub" model that assumes homogeneous water levels within each coastal segment. Our paper includes a discussion of the limitations of the bathtub approach, as well as the challenges of

implementing more sophisticated, hydrodynamic modeling approaches, in Section 3.3 Model Limitations and Planned Improvements. This text is reproduced for reference below:

> *Our reflection of local mean and extreme sea levels is limited by the resolution of our local MSL projections (1 degree in FACTS, 2 degrees in LocalizeSL) and our ESL distributions from CoDEC (50km coastline spacing). Because of the desire to build a globally consistent model using these inputs, we employ a ``local bathtub" model in which all points nearest to a given pair of ESL and MSL prediction points receive the same mean and extreme sea level projections. While this local model preserves the substantial large-scale spatial heterogeneity in SLR and ESL, sub-grid-scale variation is ignored. In particular, bathtub models are known to overestimate storm surge in inland areas largely due to the deceleration of flows caused by surface roughness (Bootsma, 2022; Vousdoukas et al., 2016). A more sophisticated, dynamic representation of ESL based on local hydrodynamic simulations for each MSL/ESL combination is beyond the computational scope of this analysis but may yield improved future results and could be incorporated either "on-the-fly" within the pyCIAM model or in a pre-processing step that updates the ESL distributions in SLIIDERS.*

To comment further on the resolution limitations, we have also added the following paragraph to that same section:

> *Future development that refines the spatial resolution of our coastal segments from the current 50 km spacing would enable a finer analysis of the local dynamics of hazard, exposure, and potential adaptation decisions for hyper-local decision making entities. At present, such an effort is hampered by a lack of more granular global inputs used to generate the SLIIDERS dataset. In particular, the 50 km-resolution CoDEC ESL dataset and the one (FACTS) or two (LocalizeSL) degree resolution of the SLR projections represent the best available observations and projections of sea levels at global scale. Using finer-resolution coastal segments would provide limited gains in model precision as the only sub-50km variation would be from enabling decision-making to occur at finer scales, rather than the incorporation of higher-resolution hazard information. The resolution at which protection and retreat decisions are and will be made likely varies substantially around the world, and thus it is unclear whether such an approach would yield more or less valid adaptation projections. As the quality and resolution of related input datasets evolve, SLIIDERS and pyCIAM can be updated to reflect these advances.*